# Finding positive meaning in memories of negative events adaptively updates memory

Megan E. Speer [1✉], Sandra Ibrahim[2], Daniela Schiller [3,4] & Mauricio R. Delgado [2✉]

Finding positive meaning in past negative memories is associated with enhanced mental health. Yet it remains unclear whether it leads to updates in the memory representation itself. Since memory can be labile after retrieval, this leaves the potential for modification whenever its reactivated. Across four experiments, we show that positively reinterpreting negative memories adaptively updates them, leading to the re-emergence of positivity at future retrieval. Focusing on the positive aspects after negative recall leads to enhanced positive emotion and changes in memory content during recollection one week later, remaining even after two months. Consistent with a reactivation-induced reconsolidation account, memory updating occurs only after a reminder and twenty four hours, but not a one hour delay. Multi-session fMRI showed adaptive updates are reflected in greater hippocampal and ventral striatal pattern dissimilarity across retrievals. This research highlights the mechanisms by which updating of maladaptive memories occurs through a positive emotion-focused strategy.

[1] Department of Psychology, Columbia University, New York, NY, US. [2] Department of Psychology, Rutgers University, Newark, NJ, US. [3] Department of Psychiatry, Icahn School of Medicine at Mount Sinai, New York, NY, US. [4] Department of Neuroscience and Friedman Brain Institute, Icahn School of Medicine at Mount Sinai, New York, NY, US. ✉email: speer.meg.e@gmail.com; delgado@psychology.rutgers.edu

We all have memories we might want to forget. Recalling negative experiences can retrigger those same painful feelings all over again—like the sting of an unexpected loss or the disappointment of a crushing failure[1]. This can be maladaptive when we ruminate about the situation—a key feature of depression and stress-related disorders[2]. Negative auto-biographical memories of high personal significance and arousal can induce a stronger sense of re-living the actual event, contributing to the retention of such memories and exacerbating clinical symptomology[3]. One potential way to alter how we feel about past adversity is to find positive meaning in it. Finding more adaptive ways to reframe negative events is central to therapeutic techniques and linked to fewer depressive symptoms, more positive emotionality[4], and faster recovery from stress[5]. However, a critical question is whether focusing on the positive aspects of past negative events actually changes the memory representation itself.

Memory is reconstructed at the time of retrieval[6], leaving the potential for modification when it is reactivated, allowing for new information to update the old through reactivation-induced updating, such as the reconsolidation process[7]. Indeed, successful updating for conditioned fear memory has been observed in rodents and humans, with similar evidence for procedural and episodic memory[8–11]. Yet, it is unclear if positive emotion-focused coping can successfully update negative autobiographical memories that we naturally recall in everyday life. Conceivably, focusing on the bright side (e.g., learning better study skills) of a past negative memory (e.g., failing an exam) could lead to the updating and re-emergence of positivity the next time the memory is retrieved, in turn lessening the experience of negative emotion at future recollections, which may also be observable in its neural representation across time. This may involve neural systems associated with updating the content and affect associated with a memory, such as hippocampus, which mediates neural reinstatement of episodic events leading to successful remembering[12], and the ventral striatum (VS) and ventromedial prefrontal cortex (VMPFC)—reward-related circuitry associated with the subjective value and positivity of recollection[13]. Thus, the present research asked whether positive meaning finding can update negative autobiographical memories with positive content, subsequently changing how we feel (emotion induced during recall), what we remember (content), and how memory is represented in the brain across time.

In this work, we show converging behavioral and neural evidence across four experiments that focusing on the positive aspects of past negative events adaptively updates memory,

leading to enhanced positive emotion and memory content at future retrieval. Adaptive updates to memory are long-lasting, remaining even after two months, and may occur through a reactivation-induced reconsolidation process. Using multi-session fMRI, this is further reflected in greater hippocampal and striatal dissimilarity across retrievals. This research highlights the mechanisms by which updating of maladaptive memories occurs through a positive emotion-focused strategy, which may promote wellbeing and resilience to adversity.

## Results

**Positive meaning finding leads to enhanced positive emotion at future retrieval.** In Experiment 1, 102 healthy individuals (35 men; $M_{age} = 20.3$; SD = 2.9) first reactivated 12 negative auto-biographical memories by writing a description and making an emotional feeling rating (How does this make you feel in the present moment? 11-point scale: $-5$ = extremely negative, 5 = extremely positive). Participants were then divided into 4 groups and had to elaborate on aspects of each memory that they found most positive (Positive group; $n = 26$; e.g., describe something you learned, something positive that occurred because of this, or how the event is meaningful to you), negative (Negative group; $n = 25$; e.g., describe what makes this memory negative or something negative that occurred because of this), or neutral (Neutral group; $n = 25$; e.g., describe the date and location). A fourth group did not elaborate and instead performed a spatial perception task focusing on whether an arrow was pointing left or right (Distraction group; $n = 26$). To test for changes over time, participants recalled, wrote descriptions, and made ratings of these same memories again 1-week later (Fig. 1). We repeated the same retrieval instructions from Recall 1 (e.g. "recall the memory naturally") to ensure consistency. This experiment (and all others reported here) were given ethical approval by the Rutgers Institutional Review Board (IRB) for Protection of Human Subjects.

We hypothesized that only the Positive group would show enhanced positive feelings and the greatest change in memory content at future retrieval, given the link between positive-emotion-focused coping and fewer depressive symptoms, more positive emotionality[4], and faster recovery from stress[5]. Content included positivity of the event (i.e., language and tone) and dissimilarity in content (i.e., change in event details) across retrievals based on two independent raters who were blind to group assignment (10-point scales; Cronbach alpha = 89.6%). Importantly, we only included memories in our analyses that were reported to be the same memory across retrievals, to rule out

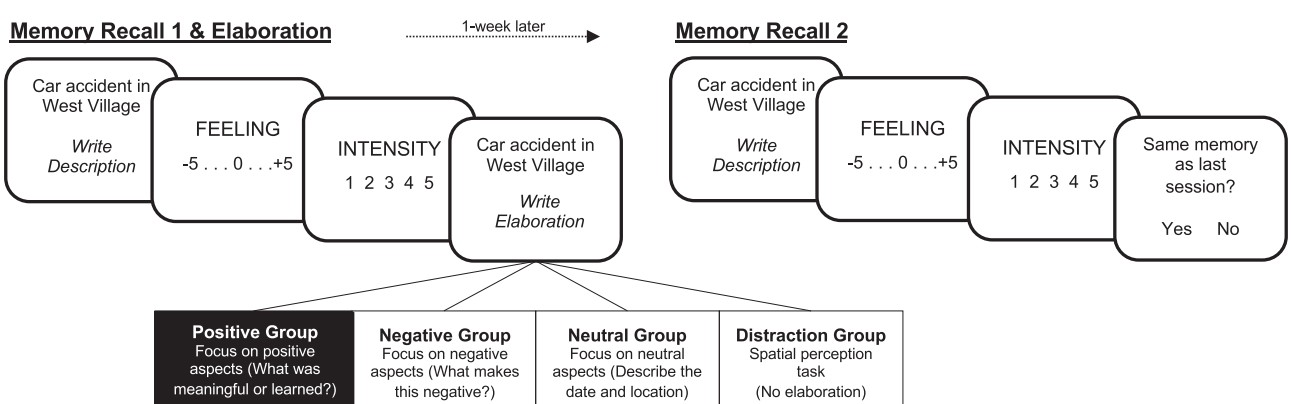

**Fig. 1 Timeline for experiment 1.** Participants first wrote about and emotionally rated 12 negative autobiographical memories. Depending on random group assignment, they either focused on the positive (Positive group), negative (Negative group), or neutral aspects of each memory (Neutral group), or performed a spatial perception task (Distraction group). To examine memory change, participants returned 1-week later to write about and emotionally rate their memories again.

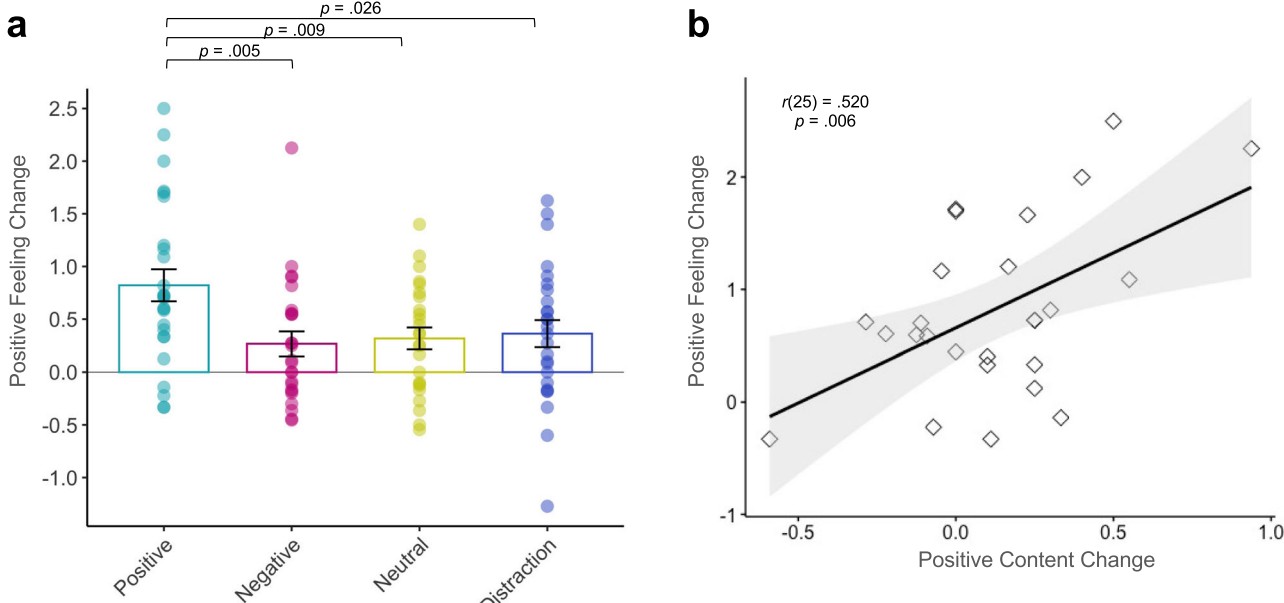

**Fig. 2 Positive emotion change across groups.** Experiment 1: **a** The Positive group ($n = 26$) showed the greatest increase in positive emotion across sessions, compared to the Negative ($n = 25$), Neutral ($n = 25$) and Distraction groups ($n = 26$). Analyses were two-tailed $t$-tests. Overlaid dots represent individual participants. Bars show mean values per group; error bars = ± SEM. **b** Within the Positive group ($n = 26$), positive emotion tracked changes in positive content over time. The shaded band represents the 95% confidence interval on the best-fitting regression line. Source data are provided as a Source Data file.

the possibility that memory change is due to retrieving a different memory in the future.

We first examined the efficacy of our regulation strategy of interest—positive meaning finding—which was indeed more positive and meaningful in content than negative or neutral elaborations (see Supplementary Information). Our key hypothesis was that finding positive meaning would elicit the greatest change in feeling across retrievals. A one-way ANOVA for feeling change (Recall2–Recall1) by group revealed a significant main effect of group, $F_{3,98} = 4.08$, $p = 0.009$, $\eta^2 = 0.111$. Post hoc $t$-tests showed that the Positive group had the greatest increase in positive emotion at future retrieval as compared to all other groups (Negative: $t_{49} = 2.87$, $p = 0.005$, $d = 0.81$; Neutral: $t_{49} = 2.72$, $p = 0.009$, $d = 0.76$; Distraction: $t_{50} = 2.30$, $p = 0.026$, $d = 0.64$), whereas the other groups did not differ from each other (all $p > 0.58$; Fig. 2a).

We also tested whether change in content across retrievals was most pronounced in the Positive group. The one-way ANOVA for change in positive content (Recall2–Recall1) was in the expected direction but did not reach significance ($F_{3,98} = 2.02$, $p = 0.116$, $\eta^2 = 0.058$) and for content dissimilarity was non-significant ($F_{3,98} = 1.53$, $p = 0.326$, $\eta^2 = 0.035$). Given our specific hypothesis regarding positive meaning finding, we performed correlations between these variables within the Positive group only. A greater increase in positive feeling across retrievals was associated with both greater increases in positive memory content ($r_{25} = 0.520$, $p = 0.006$; Fig. 2b) and dissimilarity in memory content ($r_{25} = 0.434$, $p = 0.027$). Correlations of positive feeling with positive content and dissimilarity were non-significant in the Negative ($r_{23} = 0.054$, $p = 0.799$; $r_{23} = −0.258$, $p = 0.213$), Neutral ($r_{23} = 0.281$, $p = 0.174$; $r_{23} = −0.298$, $p = 0.148$) and Distraction groups ($r_{24} = −0.010$, $p = 0.962$; $r_{24} = 0.153$, $p = 0.456$).

One potential difference between the three elaboration conditions is that positive meaning finding might inspire future-oriented thought, such as thinking about how a past negative event benefitted a future event, whereas the negative and neutral conditions might only inspire thoughts associated with the negative event itself. To test whether this possibility could explain our findings, two independent coders rated whether written elaborations included future consequences of the event. This analysis revealed no difference between the percentage of future-oriented elaborations generated by the Positive (31.2%, SD = 0.136) and Negative groups (28.6%, SD = 0.111; $t_{49} = 0.73$, $p = 0.466$, $d = 0.21$). Both groups generated more future-oriented elaborations than the Neutral group, who only described the date and location during elaboration (3.1%, SD = 0.06; Positive vs. Neutral: $t_{49} = 9.45$, $p < 0.001$, $d = 2.67$; Negative vs. Neutral: $t_{48} = 10.12$, $p < 0.001$, $d = 2.86$). Importantly, within the Positive group, future-oriented elaborations were not correlated with positive feeling change ($r_{24} = −0.11$, $p = 0.569$) or positive content change ($r_{24} = 0.028$, $p = 0.891$), suggesting that a focus on future outcomes did not drive our observed changes in memory across time.

**Changes in emotion and memory content are long-lasting.** Experiment 1 showed that positive meaning finding led to an increase in positive emotion, which tracked increased positive memory content at future retrieval. Experiment 2 aimed to replicate this finding in a larger sample and, importantly, probed the longevity of the effect over the course of 2-months. A key change in the experimental design tested the positive meaning finding strategy against natural recollection—a condition to control for memory rehearsal without the intention of modification. Ninety-one healthy individuals (39 men; $M_{age} = 20.9$; SD = 3.89) participated in a 3-session longitudinal study. The design included first and second sessions 1-week apart ($M_{days} = 7.70$, SD = 2.31) and a third session 2-months later ($M_{days} = 54.5$, SD = 6.10; Fig. 3a). Participants wrote about 10 negative memories followed by a written elaboration period of either finding positive meaning (Positive group; $n = 46$) or natural recollection (Control group; $n = 45$).

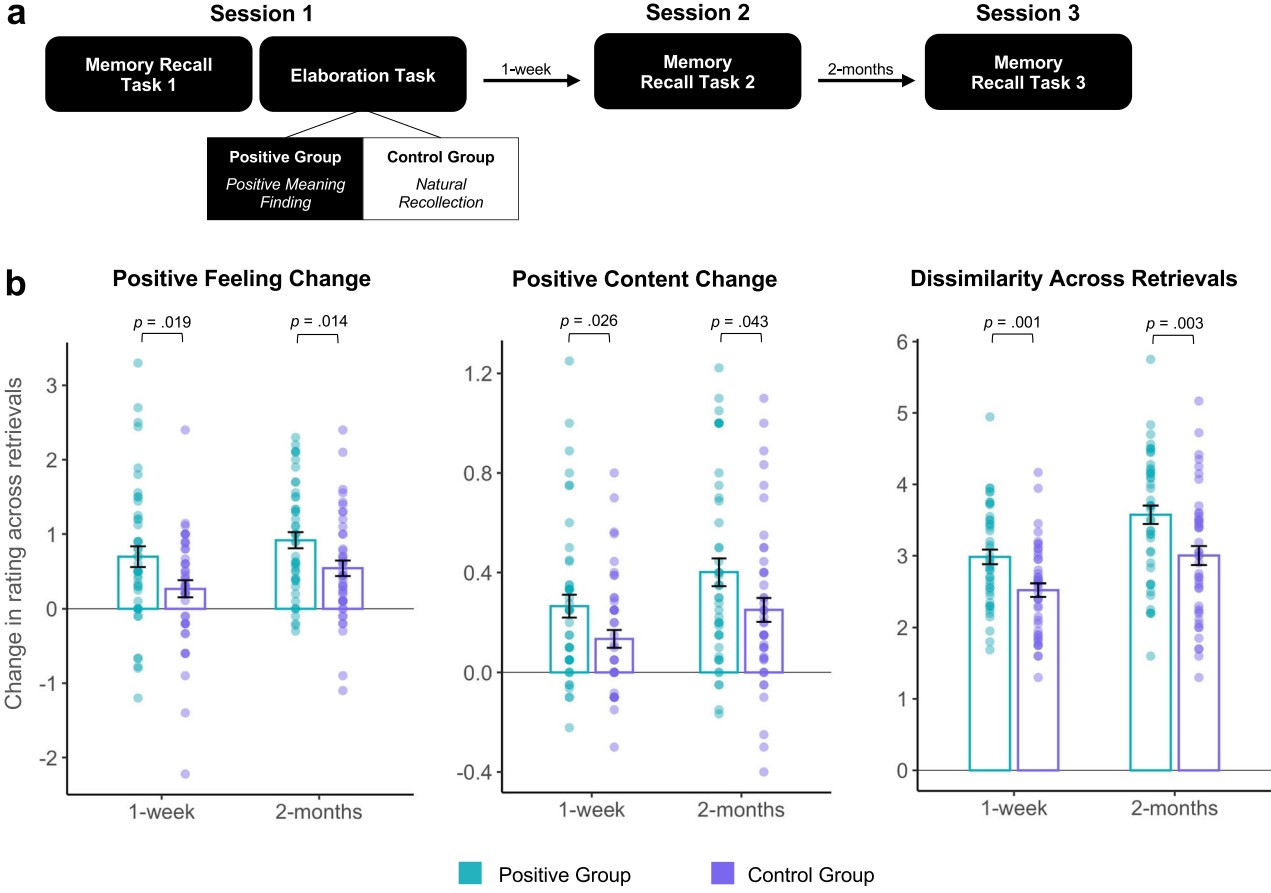

**Fig. 3 Memory change over 2-months.** Experiment 2: **a** Timeline; **b** The Positive group ($n = 46$) exhibited greater positive emotion, positive content, and more dissimilar content both 1-week and 2-months later relative to the Control group (natural recollection; $n = 45$). Analyses were two-tailed t-tests. Overlaid dots represent individual participants. Bars show mean values per group and condition; error bars = ± SEM. Source data are provided as a Source Data file.

We examined changes in feeling ratings and memory content (positivity, dissimilarity) via time (1-week, 2-months) by group (Positive, Control) ANOVAs, which yielded significant main effects of time ($F_{1,178} = 11.82$, $p < 0.001$, $\eta^2 = 0.023$; $F_{1,178} = 7.28$, $p = 0.008$, $\eta^2 = 0.037$; $F_{1,178} = 21.31$, $p < 0.001$, $\eta^2 = 0.098$) and group ($F_{1,178} = 4.43$, $p = 0.037$, $\eta^2 = .061$; $F_{1,178} = 9.10$, $p = 0.003$, $\eta^2 = 0.047$; $F_{1,178} = 19.83$, $p < 0.001$, $\eta^2 = 0.091$), but no interactions ($p = 0.824$; $p = 0.840$; $p = 0.644$). There was an increase in all three variables after 1-week and 2-months, regardless of group, potentially mirroring the fading of negative feelings that naturally occurs with time[14]. Consistent with our key hypothesis, the Positive group had a greater increase in positive emotion, positive content, and dissimilarity in event details than the Control group at both the 1-week ($t_{89} = 2.39$, $p = 0.019$, $d = 0.50$; $t_{89} = 2.26$, $p = 0.026$, $d = 0.48$; $t_{89} = 3.33$, $p = 0.001$, $d = 0.70$) and 2-month delays ($t_{89} = 2.50$, $p = 0.014$, $d = 0.53$; $t_{89} = 2.06$, $p = 0.043$, $d = 0.43$; $t_{89} = 3.07$, $p = 0.003$, $d = 0.64$; Fig. 3b). These results demonstrate that positive meaning finding has a long-lasting effect, changing the content of the memory and emotion elicited at future retrieval.

In light of these results, a key question is in what particular way does positive meaning finding alter the content of memories? To characterize the precise nature of content updating, we quantified how many event details in future recollections were from a) the initial recollection, b) integrated from the elaboration period, or c) were new related details about the event (based on ratings from 2 independent coders). Participants reported 4.74 details per memory on average ($SD = 1.36$ range = 2.3–9.0) during Recall 1, with no difference

between groups ($t_{89} = 0.568$, $p = 0.572$, $d = 0.12$). In the Positive group, a majority of details from Recall 1 were preserved across time (1-week: 74.6%; 2-months: 72.9%), which was similar to the Control group (1-week: 82.6%; 2-months: 80.3%). Groups did not differ in the number of details they recalled per memory after 1-week ($M_{Positive} = 4.19$, $SD = 1.37$; $M_{Control} = 3.50$, $SD = 2.58$; $t_{89} = 1.58$, $p = 0.117$, $d = 0.331$) or 2-months ($M_{Positive} = 4.01$, $SD = 1.22$; $M_{Control} = 4.04$, $SD = 1.10$; $t_{89} = -0.11$, $p = 0.912$, $d = 0.023$). The Positive group's future recollections included about 10.2% of their positive elaboration at 1-week and 10.5% at 2-months ($t_{45} = 4.77$, $p < 0.001$; $t_{45} = 5.27$, $p < 0.001$). Their future recollections also included 12.2% and 14.1% of new related positive details at 1-week and 2-months respectively, which was significantly greater than the Control group (1-week: 8.1%, $t_{89} = 2.63$, $p = 0.01$, $d = 0.55$; 2-months: 10.2%, $t_{89} = 2.29$, $p = 0.025$, $d = 0.48$; percentage of new details corresponds to the number of new details divided by the total number of details during retrieval). Groups did not differ in the number of new negative details that were present at 1-week ($M_{Positive} = 10.9\%$, $M_{Control} = 8.3\%$; $t_{89} = 1.74$, $p = 0.085$, $d = 0.37$) or 2-months ($M_{Positive} = 10.7\%$, $M_{Control} = 8.8\%$; $t_{89} = 1.42$, $p = 0.16$, $d = 0.30$). Given that the Positive group incorporated aspects of their positive elaboration and more new positive details than the Control group, a smaller percentage of their future recollections included initial details from Recall 1 (1-week: $M_{Positive} = 66.7\%$, $M_{Control} = 79.8\%$; $t_{89} = -3.75$, $p = 0.003$, $d = 0.788$; 2-months: $M_{Positive} = 64.7\%$, $M_{Control} = 78.4\%$; $t_{89} = -4.00$, $p = 0.0001$, $d = 0.841$). Together, these findings suggest that positive meaning finding leads to future recollections with components of both the

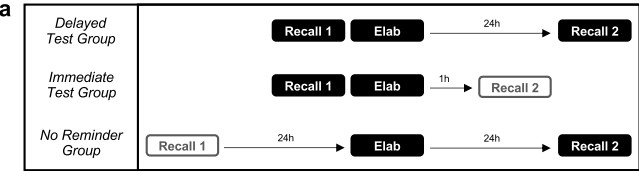

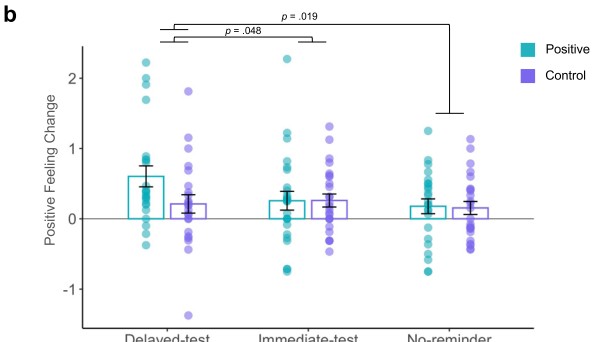

**Fig. 4 Positive meaning finding leverages reconsolidation update mechanisms.** Experiment 3: **a** Timeline; Recall 1 serves as reactivation, Elaboration is the manipulation, and Recall 2 is the future memory test. **b**) Only the Delayed-test group ($n = 23$)—individuals who underwent the positive manipulation after reactivation and were tested 24 h later— showed an increase in positive emotion for positive but not control trials at future retrieval, consistent with updating via reconsolidation. Comparison groups were the Immediate-test ($n = 25$) and No-reminder group ($n = 24$). Analyses were two-tailed $t$-tests. Overlaid dots represent individual participants. Bars show mean values per group and condition; error bars = ± SEM. Source data are provided as a Source Data file.

initial recollection and the positive elaboration, lending support to a memory updating account.

**Adaptive updating leverages memory reconsolidation mechanisms.** What are the mechanisms by which positively elaborating on a negative memory leads to beneficial changes? One intriguing hypothesis is that memory updating takes advantage of a reactivation-induced reconsolidation process. In a typical reconsolidation paradigm, memory is reactivated and then an intervention occurs during the reconsolidation window (occurring ~10 min later and up to 6 h after reactivation)[7,9]. A memory test to determine whether the memory has been modified occurs after a delay—typically 24 h—to give time for memory to restabilize. Experiment 3 follows this paradigm, whereby individuals reactivate negative memories during Recall 1 (via mental recall; 14 s) followed by the manipulation during Elaboration (i.e., positive meaning finding; 20 s). Change in memories (i.e., feeling change) are tested after a 24h-delay (Delayed-Test group; Fig. 4a).

In addition, because reconsolidation is a time-dependent process, memory updates should only be observable a) if the memories were reactivated prior to the manipulation and b) after reconsolidation has ended (after a delay) but not immediately after the manipulation[8,9]. Thus, one control group underwent the manipulation 24 h after memory reactivation, meaning outside the ~6 h updating window, and was tested after a 24 h delay (No-Reminder group). Another control group underwent the manipulation immediately after memory reactivation, meaning inside the ~6 h updating window, but was tested shortly after the manipulation (Immediate-Test group; Fig. 4a). Across all groups, half of the memories were recalled naturally as a control for comparison. We predicted that only individuals who reactivated their memories immediately prior to the positive manipulation (within the updating window) and were tested after a 24 h delay

(Delayed-Test group) should show evidence of an update in memory (as indexed by feeling change). Importantly, since the updating window is proposed to begin at least 10 min (and up to 6 h) after reactivation[15], the positive manipulation (Elaboration) occurs 10 min following reactivation (Recall 1) in the two groups where updating is meant to occur within the updating window (Immediate-Test and Delayed-Test groups).

Seventy-two healthy individuals (29 men; $M_{age} = 22.3$; SD = 6.54) were randomly assigned to three experimental groups: Delayed-Test ($n = 23$), Immediate-Test ($n = 25$), and No-Reminder ($n = 24$). When checking for baseline memory differences, we found that the No-Reminder group had significantly greater baseline feeling ratings than the Immediate-test group ($t_{47} = 2.81$, $p = 0.007$, $d = 0.80$) while neither group differed from the Delayed-test group ($t_{45} = 1.30$, $p = 0.201$, $d = 0.38$; $t_{46} = 1.52$, $p = 0.137$, $d = 0.43$). Therefore, we controlled for baseline feeling ratings in our analyses (see Supplementary Information for analyses of all baseline ratings). A condition (positive, control) by group (Delayed-Test, Immediate-Test, No-Reminder) ANOVA for feeling change, controlling for baseline feeling ratings, revealed a significant main effect of condition ($F_{1,137} = 5.32$, $p = 0.023$, $\eta^2 = 0.03$) and group ($F_{2,137} = 3.43$, $p = .035$, $\eta^2 = 0.04$), but no interaction ($F_{1,137} = 1.81$, $p = 0.181$, $\eta^2 = 0.02$). The Delayed-Test group had a significantly greater increase in positive emotion for positive relative to control trials as compared to the Immediate-Test ($t_{45} = 2.44$, $p = 0.019$, $d = 0.74$) and No-Reminder ($t_{44} = 2.03$, $p = 0.048$, $d = 0.66$) groups, who showed no such changes and also did not differ from each other ($t_{46} = 0.291$, $p = 0.772$, $d = 0.055$; Fig. 4b).

Although all three groups underwent a reminder and a positive elaboration, only one group (Delayed-Test) evinced memory updating. This finding is consistent with a reactivation-induced reconsolidation process that only a brief reminder followed by elaboration (within the ~6 h updating window) and testing after a delay would lead to long-term memory change, rather than alternative explanations, such as a short-term memory change or when elaboration is temporally distant from the reminder, which would not affect the time-limited reconsolidation process. These results show that memory modifications via positive meaning finding do not simply reflect the remembering of the last elaboration session at future retrieval—otherwise all 3 groups would have shown such changes—but rather memory is adaptively updated with positive content, having a lasting effect on future remembering.

**Positive meaning finding leads to changes in the neural representation of memory across time.** As a next step, we explored whether beneficial memory updates would also be observable across retrievals in the brain. We hypothesized that adaptive updating might be reflected in greater neural dissimilarity linked to emotion change across time in regions previously associated with memory processing and positive affect (hippocampus, striatum, VMPFC). To test this, Experiment 4 followed a modified version of the Delayed-test group in Experiment 3, including the 10 min delay between recall and elaboration. Thirty-two participants (12 men; $M_{age} = 22.8$; SD = 4.67) underwent two fMRI scans occurring 24 h apart. During scan #1, participants mentally recalled 32 negative memories and made emotion ratings (Recall 1). Afterwards (Elaboration task), they naturally recalled memories (16 control trials) or used positive meaning finding (16 positive trials). To examine changes over time, they returned 24 h later (scan #2) to recall and emotionally rate the same 32 memories again (Recall 2, Fig. 5a). They also returned for a 2-month behavioral follow-up session to examine long-term memory change.

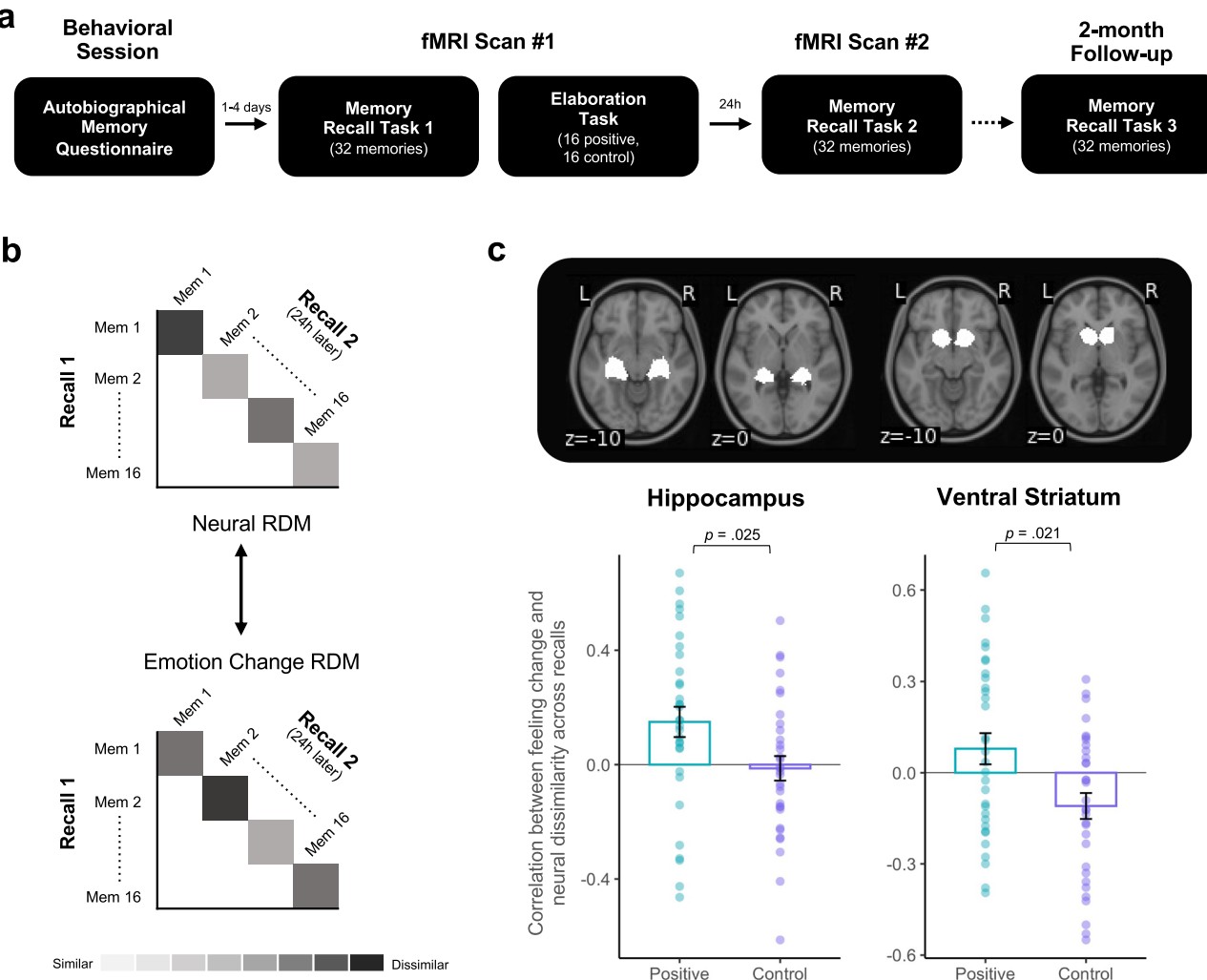

**Fig. 5 Neural pattern dissimilarity across retrievals.** Experiment 4: **a** Timeline; **b** Each neural RDM was correlated with its corresponding emotion change RDM across retrievals for each condition (positive, control). **c** We observed greater hippocampal and VS pattern dissimilarity across retrievals as a function of increasing positivity for memories that underwent positive elaboration relative to natural recall. Analyses were one-sample sign permutation tests (5000 iterations) on the difference in spearman rho values between conditions. Overlaid dots represent individual participants. Bars show mean values per condition across participants ($n = 32$); error bars = ± SEM. Source data are provided as a Source Data file.

As shown in Experiments 1–3, here too, positively reinterpreted memories elicited greater positive emotion than naturally recalled memories after a 24 h delay ($t_{31} = 5.13$, $p < 0.001$, $d = 0.91$) and 2-months later ($n = 18$; $t_{17} = 3.23$, $p = 0.005$, $d = 0.76$). Our key hypothesis was that positively reinterpreted memories (i.e., positive trials) would have greater dissimilarity in their neural activation patterns across retrievals as a function of increasing positivity in comparison to naturally recalled memories (i.e., control trials). We reasoned that Recall1-Recall2 neural dissimilarity tracking increases in positive emotion would be more meaningful than comparing across positive and control conditions alone since not all memories will necessarily change after using positive meaning finding and those that do change, change to varying degrees[16] (see Supplementary Information for exploratory analyses examining overall neural dissimilarity and exploratory ROIs, e.g., amygdala).

We tested this using representational similarity analysis (RSA) in a priori ROIs (hippocampus, VS, VMPFC). We selected the hippocampus given its role in memory retrieval[12] and the reward-related ROIs (VS and VMPFC) given their links to the subjective value and positivity of recollection[13]. We first computed a GLM modeling each memory as a single regressor for each recall period

(Recall 1, Recall 2) and participant separately. We then extracted multivariate activation patterns from each ROI, defined by a neurosynth parcellation, for each condition (positive, control) during each retrieval (Recall 1, Recall 2). We then computed the correlation distance across retrievals for a particular memory and used these values to create representational dissimilarity matrices (RDMs) for each condition. We created similar RDMs for emotion change across retrievals and correlated the neural RDM with the corresponding emotion change RDM for each condition and each participant. We then compared the mean correlations between conditions (positive, control) at the group level by running a one-sample sign permutation test (5000 iterations) on the difference in rho values between conditions using the nltools python toolbox. Consistent with our prediction, this analysis revealed greater neural dissimilarity across retrievals as a function of increasing positivity in the positive relative to the control condition, in both the hippocampus ($t_{31} = 2.36$, $p = 0.025$, $d = 0.42$) and VS ($t_{31} = 2.42$, $p = 0.021$, $d = 0.43$, Fig. 5b, c), but not the VMPFC.

In addition to our key analysis, we also performed whole-brain analyses contrasting positive and control trials during each task. During the Elaboration task, positive meaning finding (relative to

natural recollection) yielded activity consistent with prior studies examining positive reappraisal in particular (ventral striatum, caudate, and vmPFC)[17,18] and cognitive reinterpretation more generally (VLPFC, DLPFC, and DMPFC; Supplementary Fig. 2)[19,20]. Interestingly, we observed similar activation during Recall 2 for positive relative to control trials when tracking increases in positivity on a trial-by-trial basis, suggesting that updated memories may re-engage some of the same corticostriatal circuitry that was previously engaged during positive elaboration. (See Supplementary Information for whole-brain analyses of positive > control trials during the Elaboration task, Recall 1, Recall 2, Recall 2 parametrically modulated by positive feeling, parametric regression of neural activity during positive elaboration that tracks pattern dissimilarity across retrievals, and correlations between activity during Elaboration and future feeling change).

## Discussion

This research tested a potential strategy for adaptively updating negative autobiographical memories with positive content: by finding positive meaning in them. Across four experiments, we found converging evidence that positive meaning finding led to enhanced positive emotion and positive content at future retrieval. Notably, we replicated these findings four times across different experimental contexts. Our results were consistent regardless of whether people thought about or wrote about their memories, used positive meaning finding in isolation or in conjunction with natural recall, with post-retrieval memory changes after both shorter (24 h, 1-week) and longer delays (2-months), highlighting the durability and longevity of the effect. Importantly, we examined how positively elaborating on negative memories leads to memory modification, finding that it may leverage reactivation-induced reconsolidation mechanisms as updating occurred only after a reminder and 24 h delay. Such adaptive memory updating was further reflected in greater hippocampal and striatal dissimilarity across retrievals, suggesting that positive meaning finding changes how we feel, what we remember, and how memory is represented in the brain across time.

These findings join a burgeoning literature on reactivation-induced memory updating[8,9,21,22]. Here, introducing new relevant information (i.e., positive reinterpretation) after reactivating an existing memory led to an integration of the positive reinterpretation into the negative memory trace, thus modifying future recollections in a beneficial way. This conceptualization differs from alternative mechanisms that could also lead to future memory change, such as concurrent or competing retrieval of dual memories[23], which was not supported by our data (see Exp 3). That is, updates were only observable after a previous reminder during the reconsolidation window (manipulation occurring ~10 min to 6 h after reactivation) and a memory test after a sufficient delay (24 h later). Importantly, we assume there to be similar mechanisms across studies given that all other paradigms mirrored a version of the Delayed-test group in Exp 3. Similar evidence of updating has been observed across various memory domains. For instance, in associative memory, extinction after reactivating conditioned fear memories can reduce physiological arousal at future retrieval in humans[24] and freezing behavior in rodents[21] through a reconsolidation process; in procedural memory, a new list of words or methodological sequence can intrude something previously learned;[25,26] and in episodic memory, listening to others' recollections or viewing others' photographs can color one's own memory for the same event[22,27,28]. This can occur even when the new information is incorrect (e.g., misinformation effect)[29] or is a reconceptualization of what could

have been rather than what was (e.g., counterfactual thinking)[30]. Our results expand this literature to naturalistic events from one's personal past. Rather than providing external intervening information for updating, individuals internally generated this on their own, mirroring the regulatory strategies we naturally use to cope with adversity in everyday life.

An innovative aspect of the present research is examining cognitive regulation as a tool for memory updating, beyond its known role in changing our current emotional state. Decades of research highlight cognitive regulation as being highly effective in reducing negative feelings associated with an adverse event by changing how we think about it[4,19,31]. Longitudinal investigations started to hint at the idea that regulation could have a lasting impact on emotion and memory, for instance, by showing that repeated regulation training can lead to a persistent reduction in negative reactivity to the same stimuli across time[32,33]. However, these studies primarily used non-naturalistic stimuli (e.g., IAPS images or video depictions) that are less self-relevant and may rely less on memory systems than when thinking about one's own historical past[34]. Studies examining the interaction between cognitive regulation and autobiographical memory found similar reductions in negative feelings that mirror the regulation of ongoing experiences, but only focused on immediate or short-term effects (up to 30 min later)[35–38]. That finding positive meaning in past negative events can change both how we feel and what we remember in the future provides evidence of the multifaceted role and long-lasting impact of cognitive regulation strategies on psychological wellbeing.

An advantage of positive emotion-focused coping is that it is unique from other emotion regulation strategies (e.g., distancing), as it does not only dampen negative emotion. It also enhances positive emotion, leading to a greater overall emotion change, which is reflected in subjective reports, physiology[39], and differential neural responses linked to reward (striatum, VMPFC)[20], beyond cognitive regulatory regions[19]. Further, negative memories might be especially conducive to updating after using positive emotion-focused coping because a positive perspective may trigger a prediction error. Prediction error signals fuel learning and are necessary for consolidation of acquired information into a memory trace and later updating after new learning[7,40]. Although our study was not designed to test for prediction errors directly, others have done so[41], and our findings are consistent with this account.

Our RSA findings provide evidence that thinking positively about negative memories can lead to more dissimilar activation patterns at retrieval as a function of increasing positivity in regions critical to processing memory and positive affect. The hippocampus broadly contributes to encoding and retrieval, further exhibiting stable multivariate patterns across similar reactivations[42], and mediating episodic reinstatement leading to successful remembering[12]. Importantly, specific episodic details are distinguishable across multivariate patterns within the hippocampus[42], fitting with our observation that natural recollection had a similar hippocampal pattern, whereas positive meaning finding had greater pattern dissimilarity that tracked increases in positivity across time. Pattern variability in the VS linked to increased positivity is consistent with its prominent role in reward-processing and motivational states[17]. Since our fMRI design included mental rather than written recollections, future research could seek to characterize neural systems supporting emotion change versus content change in updated memories.

There are limitations to this research worth discussion. First, this strategy may not work for everyone, as individuals vary in their cognitive regulation ability. For instance, anhedonia was associated with less successful memory updating (see Supplementary Information), echoing prior research linking depression

symptomology to difficulty in positive recall[3]. Second, there may be characteristics about a memory itself that makes it less susceptible to modification. Memories with high emotional arousal, negative valence, and/or vividness, such as memories for traumatic events, might be more challenging to update. Third, we used a rating scale across a continuum from negative to positive feeling, but future paradigms may want to tease apart changes in positive emotion and negative emotion separately. This would be particularly important for testing within a reconsolidation paradigm as such studies primarily targeted negative emotion. Fourth, can other positive emotion-eliciting strategies also update memory? In a separate cohort, we found that an alternative strategy—receiving an extrinsic monetary reward after reactivation—that similarly increases positive emotion but is irrelevant to the targeted memory was less effective for updating (see Supplementary Information), suggesting that the relevance and meaningful context of the updating strategy may matter. Future research could explore the precise characteristics (e.g., relevance) and boundary conditions (e.g., duration) of manipulations leading to effective memory updating.

The desire to change how we remember our past is not new. Finding ways to lessen the deleterious impact of negative autobiographical memories has long captured the attention of researchers and is a prominent objective in therapeutic contexts[43]. The present research highlighted one such strategy. Not only do people already use positive meaning findings in their daily lives[39], but it also does not ask individuals to forget aspects of their memory or change their memory with artificial information, giving it high ecological validity. Across four experiments, focusing on the positive aspects of past adversity led to beneficial changes to long-term memory, thus capturing how memory is naturally transformed in everyday life when we choose to look on the bright side.

## Methods
### Experiment 1
*Participants.* Participants in this 2-day study included 131 healthy young adults who were randomly assigned to four experimental groups (Negative = 33, Positive = 33, Neutral = 33, Distraction = 32). Exclusions included failure to return for the second session ($N = 4$), computer issues ($N = 1$), those who did not recall specific negative memories ($N = 12$), or had fewer than 50% of memories that met criteria ($N = 12$; see Supplemental Table 1 for details on exclusions by group). The final sample included 102 participants (35 mean; Mean age = 20.3; SD = 2.9; 16.7% Asian, 22.5% Black, 23.5% Hispanic, 2.0% Pacific Islander, 41.2% White, 2.0% more than one race) across the Negative ($N = 25$; 9 men), Positive ($N = 26$; 8 men), Neutral ($N = 25$; 8 men), and Distraction groups ($N = 26$; 10 men). We conducted a power analysis using G*Power to determine sample size. We predicted a medium effect size as a conservative estimate based on similar paradigms[44], which yielded a target sample of 100 participants (25 per group; 80% power). Participants gave informed consent in accordance with the Rutgers Institutional Review Board (IRB) for Protection of Human Subjects and received partial course credit and monetary compensation for participating.

*Experimental design*

### Day 1: Memory recall 1 and elaboration.
Participants first completed questionnaires asking about their depressive and anxiety symptoms (Mood and Anxiety Symptom Questionnaire, MASQ)[45], their ability to savor positive emotions in daily life (Emotion Regulation Profile Revised; ERP-R)[46], social and positive coping strategies (COPE Inventory)[47], use of cognitive reappraisal and suppression (Emotion Regulation Questionnaire; ERQ)[4], trait resilience (Connor-Davidson Resiliency Scale; CD-RISC)[48], and recent life stress (Perceived Stress Scale; PSS)[49].

Next participants were given a list of 30 life event cues (e.g., Family Vacation) and indicated which cues triggered a specific negative memory. They then performed a memory recollection and elaboration task (Eprime 2.0 software) using 12 event cues randomly selected from that list. On each trial of this task, they saw one event cue (e.g., Witnessing an accident) and wrote 3–5 sentences describing what happened in that memory, followed by emotion ratings for feeling (How does this make you feel in the present moment? 11-point scale: −5 = extremely negative, 0 = neutral, 5 = extremely positive), intensity (How emotionally intense is this memory? 5-point scale: 1 = not intense, 5 = extremely intense), vividness (How clearly can you see this memory in your mind? 5-point scale: 1 = not vivid; 5 = extremely vivid), and age of the memory (How long ago did this occur?).

Afterwards, they wrote an additional 3–5 sentences elaborating on the memory. Depending on group assignment, they either focused on the positive aspects (Positive group), negative aspects (Negative group), or neutral aspects (Neutral group) of the memory. The distraction group did not elaborate and instead performed a spatial perception task for 30 sec after memory recall and ratings. This involved seeing an arrow and answering whether it was pointing to the left or right. This task has been used as a control condition in prior emotion regulation studies[38]. Immediately before and after the recall 1 and elaboration task, participants rated their current mood state via the Positive and Negative Affective Schedule (PANAS)[50].

### Day 2: Memory recall 2.
Participants returned 1 week later for a follow-up memory recollection test for the same 12 memories. They provided a written description (3–5 sentences) and emotionally rated (feeling, intensity) each memory again. They were also asked whether they recalled the same memory as the last session, so we could ensure that memory change was not due to recalling a different memory in the future. Like Day 1, participants rated their current mood state via the PANAS before and after the recall 2 task.

*Data analysis.* To assess change in emotion from before to after the elaboration task, we created difference scores across sessions for feeling (recall 2 – recall 1). To test for group differences in emotion change, we performed one-way ANOVAs examining feeling change by a group.

We also analyzed written memory descriptions for changes in content across time. Two independent raters made subjective ratings of positivity (1 = not at all; 10 = extremely) of descriptions from all three tasks, degree of similarity in content between retrievals (1 = extremely dissimilar; 10 = extremely similar) and ratings of meaningfulness for written elaborations (How meaningful or significant is this? 1 = not at all; 10 = extremely). Inter-rater reliability was high (cronbach's alpha = 89.6%). We then calculated a difference score for positivity across sessions (recall 2–recall 1), and performed one-way ANOVAs for change in positivity by group. Within the Positive group, we additionally performed correlations between feeling change and a) changes in positive content across retrievals and b) similarity in content between retrieval sessions.

*Exclusion criteria and data analysis common to all experiments.* In paradigms with written recollection (Experiment 1–2), we first verified that participants followed directions by having two independent raters read written descriptions from the Recall 1, Elaboration, and Recall 2 (or 3) tasks ($N = 12$ exclusions for not following directions in Exp 1). Memories were excluded from analysis if they were (a) general instead of specific, (b) not negative during the recall 1 task, or (c) if participants failed to follow directions during the elaboration task (21.2% of memories excluded). In all paradigms, participants were asked to report whether they recalled the same memory/event across retrieval periods. We only included memories that were consistent across retrievals in analyses to rule out the possibility that observed changes were due to a different memory being recalled in the future. Participants who did not have at least 50% of memories that met the criteria were excluded from analysis ($N = 12$ in Exp 1; see Supplement for counts of all exclusions in each Experiment). We followed up significant effects with post-hoc $t$-tests. All tests were two-tailed and had an alpha level of .05. In all experiments, we collected data using Eprime 2.0 software, and analyzed behavioral data using R (version 3.6).

### Experiment 2
*Participants.* One hundred and twenty-eight healthy young adults participated in this online 3-session longitudinal study and were randomly assigned to 2 groups (Positive = 65; Control = 63). Exclusions included failure to complete the second ($N = 20$) or third session ($N = 11$) or poor performance on the memory recall tasks (did not recall specific negative memories, $N = 2$; difficulty using positive meaning finding, $N = 4$). The final sample included 91 participants (39 men; Mean age = 20.9; SD = 3.89) across the Positive ($N = 46$; 20 men) and Control groups ($N = 45$; 19 men). Using G*Power, our target sample size was calculated to be 90 participants (45 per group) based on a power analysis expecting a small effect size (for detecting differences in written memory content as indicated in Experiment 1; 80% power). Participants were recruited online from the psychology subject pool at Rutgers University and gave informed consent in accordance with the Rutgers Institutional Review Board (IRB) for Protection of Human Subjects. They received partial course credit and/or monetary compensation for participating. We did not collect race/ethnicity data for this sample.

*Experimental Design.* This study mirrored Experiment 1 but was modified to include a 3rd session (2-months later) and only had two groups (Positive, Control). All sessions were conducted online using Qualtrics surveys. In session 1, participants first completed questionnaires asking about demographics, emotion, mood, and clinical symptoms—the same as in Experiment 1.

Participants were then given a list of common life event cues (e.g., Family vacation) to help them trigger 10 different, specific negative autobiographical memories from their past. For each memory they provided a specific key phrase to be used in future sessions, a 3–5 sentence description of the memory, and ratings of

feeling, intensity, vividness, significance, social closeness, frequency of recall, and date of the memory. Afterwards they elaborated further on the same memories. Depending on the random group assignment, they either wrote an additional 3–5 sentences focusing on the positive aspect of the memory (Positive group) or recalled it naturally again (Control group).

Session 2 occurred 1-week later ($M_{days} = 7.70$, SD = 2.31) and session 3 occurred 2-months later ($M_{days} = 54.5$, SD = 6.10). In sessions 2 and 3, participants saw their same 10 keyphrases, described each memory in 3–5 sentences, and made memory ratings (feeling, intensity, vividness, significance, and social closeness). In session 3, we also asked the degree to which participants talked about (in person, text/phone, or on social media), thought about, focused on the positive aspects, and focused on the negative aspects of their memories over the past 2 months.

*Data Analysis.* Data analysis procedures mirrored that of Study 1 with a few modifications. In addition to difference scores after 1-week (Recall 2 – Recall 1), we calculated an additional difference score for each dependent variable at 2-months relative to the first session (Recall 3 – Recall 1). We examined the effects of memory ratings and content across time with time (1 week, 2 months) by group (Positive, Control) ANOVAs for each dependent variable. Memory content was judged by two independent raters on the same characteristics of positivity and dissimilarity. Inter-rater reliability was high (cronbach's alpha = 94.9%).

### Experiment 3
*Participants.* One hundred and four healthy young adults participated in this multi-session study and were randomly assigned to 3 experimental groups (Immediate-test = 35, Delayed-test = 35, No-reminder = 34). Exclusions included failure to return for the second or third session (N = 11, N = 1; due to adverse weather), computer issues (N = 3), and poor performance on the memory recall tasks (did not recall specific negative memories, N = 4; remembered <50% of memories from Day 1, N = 6; difficulty using positive meaning finding, N = 7). The final sample includes 72 participants (29 men; Mean age = 22.3; SD = 6.54; 6.4% Asian, 31.9% Black, 26.4% Hispanic, 23.7% White) across the Immediate-test (N = 25; 10 men), Delayed-test (N = 23; 10 men), and No-reminder groups (N = 24; 9 men). We used G*Power to determine sample size. We expected a large effect size (based on Experiment 1 results for emotion change across groups), which yielded a target sample size of 75 participants (25 per group; 80% power). Participants gave informed consent in accordance with the Rutgers Institutional Review Board (IRB) for Protection of Human Subjects and received partial course credit for participating.

*Experimental design*

### Day 1: Autobiographical memory questionnaire.
This session followed the same paradigm as Exp 1–2 but with a few modifications. Their AMQ included 68 negative event cues (e.g., Witnessing an accident). For each cue, they thought of a negative memory, wrote a brief description (1–2 sentences), provided a date, and made subjective ratings of feeling, emotional intensity, vividness, significance, and social closeness. Importantly, participants also created a unique keyphrase (5–10 words) for each memory to facilitate recollection at future retrievals. The experimenter randomly selected 32 memories from each participant's AMQ that were deemed as negative and occurred at a specific place and time (the minimum criteria for inclusion). Memories were then matched in feeling and intensity ratings and each pair was randomly assigned to the positive or control condition to ensure that memories were similar in each condition at baseline.

### Day 2–3: Memory recall 1, elaboration, and memory recall 2.
All three groups of participants retrieved 32 negative memories via mental recall as a means of reactivation (Recall 1 task). On each trial, they saw one unique keyphrase for 14 s, made button presses indicating recall duration, and made ratings of feeling and emotional intensity.

Afterwards they performed an elaboration task that included the positive manipulation. On each trial, they saw the same unique keyphrases again for 20 s. They were asked to find positive meaning in the negative memory for half of the memories (16 positive trials) and to naturally recall the other half (16 control trials). After a positive trial, they were asked: Were you able to think of something positive associated with this memory? (Yes/No). After a control trial, they were asked: Did you think about the specified memory? (Yes/No). There were 2 positive blocks and 2 control blocks with 8 trials each. Blocks were presented in counterbalanced order across participants. Two groups (Immediate-test and Delayed-test) reactivated the memories prior to the positive elaboration manipulation (during the reconsolidation window). The third group (No Reminder) did not reactivate the memories immediately beforehand, instead there was a 24 h delay between reactivation and elaboration.

To assess changes in memory, all participants returned to recall the same 32 memories again via mental recall and make subjective emotion ratings (recall 2 task). On each trial, they were also asked whether they recalled the same memory as in the prior session. Two groups (Delayed-test and No-reminder) returned for recall 2 after a 24 h delay, whereas the third group (Immediate-test) returned after only a 1 h delay (during the reconsolidation updating window).

*Data Analysis.* To assess change in emotion from before to after the elaboration task, we created difference scores (recall 2–recall 1) across sessions for feeling ratings for each condition (positive, control) separately. We then performed a condition (2: positive, control) by group (3: Delayed-test, Immediate-test, No-reminder) ANOVA for feeling change.

### Experiment 4
*Participants.* Forty healthy young adults participated in this multi-session fMRI study. Exclusions included poor performance on the memory recall tasks (difficulty using positive meaning finding, N = 4; remembering fewer than 50% of memories across sessions, N = 1) and motion >3 mm in any direction (N = 3). The final sample included 32 participants (12 men; $M_{age} = 22.8$; SD = 4.67; 37.5% Asian, 21.9% Black, 18.8% Hispanic, 34.4% White). Our target sample size was 35 participants based on prior fMRI studies using similar multivariate analyses and behavioral data from Experiment 3. Participants gave informed consent in accordance with the Rutgers Institutional Review Board (IRB) for Protection of Human Subjects and received monetary compensation for participating.

*Experimental design.* This was a 4-session study occurring over the course of 2 months. It included an initial behavioral session, two fMRI sessions, and a 2-month behavioral follow-up. It mirrored the paradigm for the Delayed-test group in Experiment 3.

### Behavioral session.
In the first session, participants completed the same questionnaires (about emotion and clinical symptomology) and AMQ as described in Experiment 3. The same procedure was used to select 32 memories total (16 positive, 16 control).

### Scan 1: Memory recall 1 and positive elaboration.
In the second session (1–4 days later), participants completed two tasks while undergoing fMRI scanning. They first performed a memory recollection task in which they mentally recalled 32 negative memories (Recall 1). They saw one unique keyphrase for 14 s, thought about the memory, and made button presses indicating recall duration. After a short delay (2–4 s), they made ratings of feeling and emotional intensity (unrestricted time limit), followed by a 6–10 s ITI.

After Recall 1, participants performed a Memory Elaboration task that included positive manipulation. On each trial, they saw one of the 32 unique keyphrases again for 20 s. They were asked to positively elaborate (i.e., find positive meaning) on half of them (16 positive trials) and to naturally recall the other half (16 control trials). They made button presses to indicate a duration of recall or elaboration, followed by a 2–4 s delay. After a positive trial, they were asked: Were you able to think of something positive associated with this memory? (Yes/No). After a control trial, they were asked: Did you think about the specified memory? (Yes/No; unrestricted time limit for ratings). An ITI of 6–10 s separated one trial from the next. There were 2 positive blocks and 2 control blocks with 8 trials each. Blocks were presented in counterbalanced order across participants.

At the end of Scan 1 participants gave subjective reports of their overall performance on the task, such as their success in using positive meaning finding and natural recall (both 7-point scales: 1 = not successful at all; 7 = extremely successful), and how difficult it was to switch between strategies (7-point scale: 1 = not difficult at all; 7 = extremely difficult).

### Scan 2: Memory recall 2.
In the third session (1-day later), participants underwent their second scan and performed the same memory recall task (Recall 2; 14 s recall followed by emotion ratings). Importantly, at the end of each trial they reported whether it was the same memory as the prior session (Yes/No). This second fMRI session allowed us to measure changes in emotion and the neural representation of memories from before (Recall 1) to after positive elaboration (Recall 2).

### Two Month behavioral follow-up.
In the fourth session, (2-months later), participants performed the same memory recollection task again (Recall 3; 14 s recall followed by emotion ratings). This allowed us to test for longer-lasting changes in the memories over time and served to replicate our longitudinal behavioral study (Experiment 2).

*fMRI data acquisition and preprocessing.* We collected neuroimaging data using a 3 T Siemens Magnetom Trio scanner. We acquired structural images using a T1-weighted MPRAGE sequence in 176 1 mm sagittal slices (256×256 matrix, FOV = 256 mm) and functional images in 35 contiguous oblique-axial slices (3x3x3mm voxels) set parallel to the AC-PC plane with a single shot gradient-echo EPI sequence (TR = 2 s, TE = 25 ms, FOV = 192, flip angle 90, bandwidth = 2232 Hz/Px, echo spacing = 0.51).

Images were preprocessed using SPM 12. We motion-corrected each time series to its first volume, and then performed spatial unwarping to minimize geometric distortions due to susceptibility artifacts[51]. We coregistered the mean functional

image to the anatomical scan, normalized the anatomical using the unified segmentation model[52], and then resliced the functional data to Montreal Neurological Institute (MNI) standard stereotaxic space. We then applied spatial smoothing using a Gaussian kernel of 5 mm FWHM.

To minimize the impact of head motion, we applied additional preprocessing steps using FSL 6.0. Specifically, we detected motion spikes using the FSL tool *fsl_motion_outliers*. This tool evaluates motion spikes with two metrics: (1) root-mean-square (RMS) intensity difference of each volume relative to the reference volume from the first time point; and (2) frame-wise displacements, which are the mean RMS change in rotation/translation parameters relative to the reference volume. We labeled volumes as spikes using a boxplot threshold (75th percentile plus 1.5 times the interquartile range) for metric values within a run, and then removed these spikes via regression[53,54]. Across all participants, this removed 10% of volumes (range: 3.3 to 19.5%). We then extracted brain material from the functional images and normalized the entire 4D dataset using a single scaling factor (grand-mean intensity scaling). To remove low-frequency drift in the MR signal, we applied a high-pass temporal filter (100-s cutoff).

*fMRI data analysis.* Functional data were analyzed using a whole-brain random-effects general linear model (GLM) in FSL. Regressors were convolved with a canonical double-gamma hemodynamic response function and 24 regressors for motion parameters were included in the model. The 24 motion-related regressors included the 6 motion realignment parameters (3 orientations, 3 rotations) and their derivatives, and each of the squares of these regressors (e.g., [53]). To correct for multiple comparisons, we used a non-parametric permutation test (5000 iterations) to obtain an alpha < .05. Permutation tests provide similar results to standard FWE correction based on random field theory[55]. We performed contrasts of positive relative to control trials during the Elaboration task, Recall 1, Recall 2, Recall 2 parametrically modulated by positive feeling, and a parametric regression of neural activity during positive elaboration that tracks pattern dissimilarity across retrievals (See Supplementary Information).

*Representational similarity analysis (RSA).* We performed representational similarity analyses (RSA) to examine neural pattern similarity between the two retrieval sessions (Recall 1 and Recall 2) as a function of condition (positive, control). RSA requires the construction of representational dissimilarity matrices (RDMs), which summarize the pairwise dissimilarities between stimuli[56]. In our case, we were interested in comparing neural patterns in Recall 1 to Recall 2 in ROIs previously implicated in memory and positive affect (hippocampus, ventral striatum, VMPFC)[13,17,18,42,57].

To conduct RSA, we first computed a GLM modeling each memory as a single regressor for each recall period (Recall 1, Recall 2) and participant separately. Participants' button presses during the task defined onset and durations of recollection (see Supplementary Information for RT details). We then extracted the multivariate neural pattern in each ROI (i.e., parameter estimates) for each memory in both recall tasks for each participant. This analysis only included memories that met criteria (as described in the exclusion criteria section and common to all experiments; 23.1% of trials were excluded). After, we calculated the correlation distance between the two neural patterns of activity for Recall 1 and Recall 2 of the same memory in a particular ROI, which made up the RDM for brain space. We then constructed a similar RDM for feature space, which included the corresponding emotion change values for each pairwise comparison. To test our hypothesis that positive meaning finding leads to greater neural dissimilarity (than natural recall) across retrieval sessions, we computed RDMs separately for positive and control trials. Then, we calculated the spearman rho correlation between RDMs for brain space (in a particular ROI) with RDMs for feature space (change in feeling rating across retrievals) for each participant and for each condition, separately. Finally, we compared the mean correlation coefficients across conditions (at the group level) using a one-sample sign permutation test (5000 iterations) on the difference in rho values between conditions using the *nltools* python 3 toolbox. We chose to apply a 5 mm smoothing kernel to our data because our ROIs were sufficiently large such that smoothing may improve the signal-to-noise ratio[58].

**Reporting summary**. Further information on research design is available in the Nature Research Reporting Summary linked to this article.

## Data availability
The behavioral data that support the findings of these studies are available on OSF (https://osf.io/jtgfk/). We are unable to share written memory content because it includes identifiable information. Source data are provided with this paper.

## Code availability
The code that supports the findings of these studies is available on OSF (https://osf.io/jtgfk/).

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

## Acknowledgements

The authors would like to thank Fernanda Bonda, Shawn Fagan, Bernadette Garcia, Vivien Garcia, Madhuri Kashyap and Daniella Mendez for research assistance, and Luke Chang for assistance with neuroimaging analyses. This research was supported by the National Institute on Drug Abuse (DA027764) and the McKnight Foundation to M.R.D. and an APF F.J. McGuigan Dissertation Award to M.E.S.

## Author contributions

M.E.S and M.R.D developed the study concept. M.E.S, D.S., and M.R.D. contributed to study design. M.E.S and S.I. collected data. M.E.S analyzed data under the direction of D.S. and M.R.D. M.E.S drafted the manuscript. D.S and M.R.D. provided critical feedback and approved the final version of the paper.

## Competing interests

The authors declare no competing interests.
