## [Peer Review File · Nature Communications]

Finding positive meaning in memories of negative events adaptively updates memoryREVIEWER COMMENTS

Reviewer #1 (Remarks to the Author):

In this paper, the authors outline four well-crafted experiments examining how finding positive meaning in negative autobiographical memories may utilize memory update mechanisms. Through their experiments, they show that memories that are manipulated only during a labile period are modified, and that changes persist for up to two months after modification.

I believe this paper is acceptable with minor revision. I list some areas of clarification below.

Comments:

- There was a brief comment on the intersection of memory updating and emotion regulation in the discussion, but the experiments in this manuscript may benefit from being framed in the context of these existing studies. See Samide & Ritchey (2020): Reframing the Past: Role of Memory Processes in Emotion Regulation for a review, and Parikh, McGovern, & LaBar (2019): Spatial distancing reduces emotional arousal to reactivated memories for a recent empirical article.
- In Experiment 1, you report a correlation between positive feeling change and content dissimilarity from recall 1 to recall 2. Is this change fully explained by new positive content incorporated into the memory descriptions? If not, can you discuss the ethical and therapeutic implications of altering the content of recalled memories?
- Are you able to report full demographics of the participants in the study (e.g., ethnic/racial breakdowns?)
- Even within the restriction of negative, specific memories, memory quality and specific valence can vary greatly from individual to individual. For this reason, have you considered running the analyses as mixed models instead of ANOVAs with a fixed effect of participant? This analysis will also nicely mirror the set-up of the RSA, with each memory as an individual data entry.
- Page 18: In Experiment 1, what were the reasons to base the power analyses off of a large effect? Was this purely a conservative choice?
- Page 21: What percentage of memories were removed due to your exclusionary criteria? I ask this because Experiments 1 and 2 already have very few memories per condition per person.
- Similarly, can you reiterate how many participants were excluded in Experiment 1 due to the exclusionary criteria on page 21 (within the exclusion criteria section)? Though these numbers are reported in the participants section, listing them side by side with the exclusionary criteria would make it easier to quickly perceive whether participants were generally compliant.
- In Experiment 3, you collected a date for each memory to ensure each memory had a specific time and place. Are these dates precise enough to use as a measure of time since event? With evidence of episodic memories becoming more semantic/crystallized over time, one interesting addition to this work would be to see if more recent memories are more affected by the positivity manipulation than older memories.
- Page 28: Could you clarify which preprocessing steps were completed in SPM12? It looks as though the first paragraph describes SPM12 preprocessing steps while the additional preprocessing (e.g., motion spikes) were in FSL, but the former is not explicitly stated.
- Page 29: The whole brain fMRI analyses are set up here, but there is no explicit mention in the methods that the full analyses are in the Supplement. Furthermore, could you include a list of contrasts of interest from the whole brain GLM in the main text for additional clarity?

Reviewer #2 (Remarks to the Author):

Overall, I thought that this was a novel, creative, and meticulous paper that reported on some interesting findings. I definitely think that this paper should be published and am certain that it will make a solid contribution to the literature. I just had a few follow-up questions and suggestions, hopefully to improve the paper.

1. I thought that one of the most novel parts of the paper was the integration of a reconsolidation account. However, this seemed to be a bit buried in the paper. Thus, the introduction moves very quickly through the reconsolidation literature and the motivation for these studies. Only one sentence is spent describing the reconsolidation literature (beginning of second paragraph), which is central to the hypotheses presented. I would suggest adding a bit more to the introduction to review the relevant points from the reconsolidation literature. I would also suggest mentioning reconsolidation in the title.

2. Along the lines of reconsolidation, can the authors clarify the time passed between the Recall (reminder) and Elaboration (intervention) tasks for these studies, and how this speaks to the reconsolidation literature (whether the reconsolidation window is "open" during elaboration)? My understanding is that at least 10 minutes should pass between the reminder cue and memory-updating intervention: Monfils M. H., Cowansage K. K., Klann E. & LeDoux J. E. Extinction-reconsolidation boundaries: key to persistent attenuation of fear memories. *Science* 324, 951–955 (2009). It would be helpful to have a bit more clarity on this.

3. Did the authors differentiate between negative feeling change and positive feeling change? While it may be the case that an increase in positive feelings corresponds to a decrease in negative feelings, it is also possible that there is no change in negative feelings associated with the autobiographical memory alongside a significant change in positive feelings. This seems like an interesting and important consideration in explaining how these memories are updated, and is relevant in terms of situating this work in the reconsolidation literature since it seems like a vast majority (but not all) of reconsolidation paradigms focused on attenuating the negative affect associated with certain stimuli, whereas this paper focuses on increasing positive affect associated with particular stimuli. It looks like the scale used to assess participants' feelings ranges from negative to neutral to positive, so could the change in positive feelings described in the results be due to a decrease in negative affect as opposed to an increase in positive affect? It would help if you could methodologically clarify how this scale was used, and also discuss the theoretical implications associated with this focus on increasing positive affect versus decreasing negative affect associated with negative autobiographical memories and how this relates to other reconsolidation work.

4. While I very much appreciated the 'dissimilarity' analyses, I also found myself wondering if subjects in the positive reinterpretation condition showed general increases in ventral striatum and VMPFC activity during scan #2 relative to scan #1 (in the recall period). I later saw that these analyses were included in the supplementary materials. I would suggest considering moving these results to the main text, as my guess is that other readers will have these same questions.

Some additional minor issues:

5. Figure 1a: The last panel in "Memory Recall 2" says: "Same memory as last session?" Is this question asked to participants, or determined by coders? This figure makes it seem like the former (which seems odd to ask), but the paper description makes it seem like the latter.

6. Exclusions for Experiments 1-3 by Group: The third category (remembered <50% of memories) is confusing. Can this be explained differently?

7. Supplemental Methods & Results for Experiment 1 – Positivity of Content Rating: The description suggests that the rating is based on negativity of the description, whereas the scale is based on similarity of the description. Can you clarify this discrepancy? Small typo on this scale: "Very Dissimilarity"

8. Can you clarify on page 4 of the supplement whether Recall 1 and Recall 2 are both from the "Write description" part of the tasks (and not the "Write elaboration" part)?

9. Experiment 2 supplement: Can you include the prompt for the natural recollection group (i.e., control condition)?

10. It might help to add some information about the reconsolidation window to Figure 3A, and to clarify that Recall 1 is serving as the "reminder cue" before the window is opened, Elaboration is introducing a manipulation (hopefully during the reconsolidation window), and Recall 2 is testing whether the memory was updated. As is, it takes some effort to puzzle through the conditions introduced in the figure.

11. Can you provide the Autobiographical Memory Questionnaire you used in your supplemental materials so that readers can get a complete sense of the negative event cues?

12. Thank you for making your data available on OSF, but it does not appear to be publically available. Can you share access with reviewers? Can you additionally share the code used to run your analyses in your repository?

Reviewer #3 (Remarks to the Author):

This study used a series of behavioral experiments and an fMRI experiment to test whether focusing on positive aspects of a past negative event can adaptively update the negative memory with more positive feelings and content, ultimately changing how the memory is represented in the brain. The authors report that a positive elaboration manipulation elicited greater increases in positive feelings than negative, neutral and distraction manipulations after 1 week (behavioral Exp 1). The positive elaboration manipulation also yielded greater increases in positive feelings, positive content and memory dissimilarity after 1-week compared to a natural recollection condition, and the effect persisted for 2-months (behavioral Exp 2). In behavioral Exp 3, they report findings consistent with a reconsolidation-based account of the memory updating mechanism. Using fMRI (Exp 4), they examine whether the neural representation of memories becomes more dissimilar across recalls following positive elaboration. Although the positive elaboration and control condition do not yield overall differences in representational dissimilarity, the authors do find evidence that changes in positive emotion are correlated with neural dissimilarity across memories in the hippocampus and ventral striatum.

I am impressed with the authors' commitment to replicating their results across multiple experiments. These findings advance our understanding of a technique for updating negative autobiographical memories in a relatively naturalistic way and provide insight into the underlying mechanism. The methods appear to be sound. I have a few questions and suggestions:

1. It might be informative to specifically test if the relationship between neural dissimilarity and change in feeling rating is indeed memory-specific. For example, you could shuffle the pair assignments of neural dissimilarity and feeling rating and recompute the spearman correlation X number of times in each participant (within condition). If the relationship is indeed item-specific, then the observed value should be larger than the permuted distribution. For statistical analysis, you can compute the z-score of the true value relative to the mean and SD of the permuted distribution in each subject. Then examine the reliability of these z-scores across participants with a one-sample t test against zero.

2. In Exp 4, is there a way to test whether the recall 2 memories do indeed integrate more "positivity" in their neural representation, which would account for the increased dissimilarity from recall 1 to recall 2 (specifically in the positive relative to control condition)? This might not be possible in this particular design, but one idea is maybe you could use the fMRI data from the elaboration task to estimate an average "template" of positive emotion during recollection (perhaps by simply averaging across positive elaboration trials or contrasting positive > natural), then regressing out this template from recall 2 patterns (both positive and control trials) and re-

running the neural dissimilarity/emotion change RDM correlations.

3. Minor points:

(page 6) Please remove the word "trending" when describing a statistical test where $p > 0.1$.

(page 8) I'm noticing the df (241) reported for the correlations of future-oriented elaborations with positive feeling change and positive content change. Is this correct? Do these correlations include multiple data points per subject? If so, you should eliminate the dependencies in the correlation.

(page 8) Please report the statistics for the time x group ANOVA main effects and interaction in the main text (currently they are in the supplement). Also, the t-tests reported in the main text and supplement are redundant. I also noticed that some of the Experiment 3 and 4 results from the main text are repeated in the supplement.

(page 9) "Participants reported 4.74 details per memory on average (SD=1.36 range= 2.3-9.0), with no difference between groups" Is this during Recall 1 or all recalls together?

(page 10) How are the "new related positive detail" percentages calculated (percentages of what)? Are new related positive details at 2 months new relative to session 1 or new relative to session 2? Are you also able to report the percentage of new negative details? The authors report that the groups did not differ in the number of new negative details, but I am also curious about how the number of new negative details compares to the number of new positive details.

(page 10) "Together, these findings suggest that positive meaning finding leads to future recollections with components of both the initial recollection and the positive elaboration, lending support to a memory updating account." Do the results also indicate that they produce new positive details too (that weren't part of the initial recall or positive elaboration)?

(page 11) Please report the condition x group interaction.

(Page 13) "Our key hypothesis was that positively reinterpreted memories (i.e., positive trials) would have greater dissimilarity in their neural activation patterns across retrievals in comparison to naturally recalled memories (i.e., control trials)." My understanding is that this isn't exactly the key hypothesis that is then tested and reported in the subsequent part of the paragraph. You aren't testing for the difference in neural dissimilarity between positive trials and control trials, but rather the difference in the relationship between neural dissimilarity and emotion change between groups.

(methods - Exp 1) During session 2, were subjects specifically instructed to recall the same memory for each cue (e.g. family vacation) as they did during session 1? Or were they just shown the cue and they could choose to recall a different memory?

Supplementary info page 3 - The scale listed under positivity content rating is not correct. For the dissimilarity rating scale, I believe it should read "Very Dissimilar" not "Very Dissimilarity"

Fig S1 - when you say collapsing across 1-week and 2-month retrievals, does this mean that there are two data points per subject included in the correlation? I would prefer seeing this plot and correlation coefficient separated for 1-week and 2-month retrievals.

There are some typos in second to last sentence on page 16 of supplement.

Reviewer #4 (Remarks to the Author):

Review of "Finding positive meaning in past negative events adaptively updates memory"

This paper offers evidence to the effect that revisiting memories of past negative experiences

within the content of finding positive meaning to them, updates the underlying memory representation in adaptive ways. It is a timely paper, and the result could potentially help to illuminate the cognitive and neural mechanisms underlying behavior change in therapeutic contexts. That being said, I have a number of concerns that prevent me from recommending this manuscript for publication in its current form. I elaborate on these concerns, below.

p. 5. I found the instructions to the “meaning making” manipulation a bit open ended. It seems to me that “describe something you learned”, “think about something positive that occurred as a result of this”, and “how the event is meaningful to you”, are rather different elaboration strategies that likely bring about distinct cognitive processes and emotions. Thinking about something that occurred as a result of a past event involves causal reasoning and likely the recollection of a subsequent episodic autobiographical memory, and we do not know if that memory is positive or negative. By contrast, thinking about how the event is meaningful to oneself may not require causal reasoning, or the retrieval of a subsequent memory, but rather abstraction or some other kind of conceptual operation. My concern is that there are a number of different kinds of processes involved here, and it is not clear which one in particular may be doing the work. Notice, incidentally, that the instructions for the other two conditions are less open ended—at least from the description in the paper.

p. 5. Last two lines. What is the rationale behind this hypothesis?

p. 5. (Also, p. 21, in methods). If I understand correctly the exclusions reported in the Supplemental Information, your ns per group are smaller than what reported in page 6. Specifically, the n for the positive group is 19 (26 – 7), for the negative group is 17 (25 – 8), for the neutral group is 17 (25 – 8) and for the distraction group is 20 (26 – 6). Is this correct? Additionally, you mentioned that you excluded those participants that remembered 50% or less of the same memories in S2 relative to S1. I had to dig into the supplemental information to learn that, on average, it looks like participants contributed 9.46 memories, although we don’t know how many per group—all we know is that they did not differ statistically from one another. I think it may be important to frontload some of that information. I started reading experiment 1 thinking that each participant contributed on average 300 observations (25 x 12), but it turns out that they are effectively contributing a bit over half of that (~170). Alternatively, I may be wrong, and you are reporting ns *after* exclusions in p.5. I apologize if I misunderstood.

p. 8. Regarding experiment 2, were there any differences in the initial emotion of the reported memories (i.e., in session 1)? I could not find this information in the SI, and I think it is important that there aren’t baseline differences in affect between the groups to fully compare the effects of the manipulation. I noticed that the researchers also collected vividness, significance, social closeness, and frequency of recall. Were there any differences between groups for these measures? May be important to consider them as potential covariates.

p. 11. Experiment 3. According to the authors, in the no-reminder group there is no reactivation of the memory prior to the manipulation. But how can participants engage in thinking about positive aspects of a memory if they are not recalling it? It seems to me that for the elaboration to work in the “no reminder group”, participants still need to reactivate the memory they are going to think about, in which case it looks as though what happens in that group shouldn’t be different from what happens in the “delayed test group”. Yet, the authors find differences. I guess it may be worth clarifying how participants can find positive meaning in a remembered event without first remembering the event in which they are supposed to find meaning.

Another point about Exp. 3. Participants gave ratings of feeling, emotional intensity, vividness, etc. Do memories in session 1 differ along any of these ratings? If so, it may be important to include as regressors in the analysis, as it could rule out potential confounds.

p. Regarding experiment 4, it would be nice to know the rationale for picking those ROIs. I was surprised, for instance, not to see the amygdala included among the ROIs, given its role in emotion (it is included in the SI, but not in the main text, not sure why). I was also surprised not to see an effect for VMPFC, which has been strongly associated with self-referential processes. My guess is that the authors included it as an ROI, in part, because of this. What do you make of this

null result (p. 14)?

Minor comments:

p. 4. Second paragraph, line 2. What do you mean by "reopens"? Maybe "reactivates"?

p. 6. Third paragraph, line 3. I don't think that the result of that first ANOVA should be taken as "trending".

Reviewer #5 (Remarks to the Author):

This manuscript reports 4 experiments (3 behavioural, 1 behavioural + fMRI) examining how positive re-evaluation changes the content of originally negative autobiographical memories. All experiments show robust effects on behaviour, such that positive reinterpretation immediately following memory reactivation leads to a lasting increase in positive emotional content. Overall, this is a well-conducted set of experiments with highly coherent results.

The editors specifically asked me to comment on the RSA approach in Experiment 4 (fMRI). Here, the authors test the key hypothesis that positive re-evaluation causes a change in a memory's neural representation, such that the neural patterns become more dissimilar when comparing the original Recall1 with the delayed Recall2. To test this hypothesis, participants are scanned twice, 24h apart, while they recall and rate the emotionality of 32 memories. Immediately following Recall1, participants positively re-evaluate half of the memories, and naturally recall the other half. Single-trial multivariate patterns are extracted during Recall1 and Recall2 from the hippocampus, ventral striatum (VS) and ventromedial prefrontal cortex (VMPFC), and correlation distance (1-corr) is used as a measure of Recall1-Recall2 neural pattern dissimilarity. This is done separately for the 16 positive and the 16 control trials, creating one representational dissimilarity matrix (RDM, or rather a vector along the diagonal) per condition per subject. As reported in the supplements, the authors do not find evidence for a significantly greater pattern change after positive re-evaluation than after natural recall. However, the change in neural similarity scales (positively) with the change in emotion rating in hippocampus and VS, as shown by a second-order analysis correlating the neural pattern change vector with a behavioural emotion change vector. More specifically, the correlation between neural change and emotion change is significantly larger for positive than control trials.

The null result in the positive reinterpretation condition compared to the control condition is problematic and makes it difficult to interpret the differential second-order correlations. Surely, if the argument is that the behavioural changes shown in Experiments 1-4 are produced by a change in neural memory traces, then we would expect positive trials to show a greater on-average pattern change than control trials. Without such an overall increase in dissimilarity, it is unclear what is driving the differential correlations with emotion change, especially given that the differences are statistically quite small, and the correlations are based on only 16 trials per participant and might thus be unstable. To make this a more constructive comment, maybe the authors could at least present evidence that those memories that show the most pronounced emotion change (e.g. based on a split-half) do exhibit a significant neural pattern change from Recall1 to Recall2. Also, is it possible that the authors were looking for such pattern changes in the wrong set of brain regions? A whole-brain searchlight using a model matrix of larger pattern change in the positive than the control condition could answer this question.

The results section is also missing statistics to test if the second-order correlation is significant in the positive condition itself (i.e., when not comparing it to the control condition). From Fig. 4 it appears that in the hippocampus, the differential correlation between conditions is produced by the expected positive correlation in the positive condition; however, in the striatum, the difference is mainly produced by a negative correlation in the control condition (see Fig. 4), suggesting that it is the memories with little emotion change that become more dissimilar (or vice versa). Again, the differences between conditions seem meaningless without an overall pattern change, and a significant positive correlation, in the positive condition.

Finally, the univariate differences (and their absence in some contrasts), reported in the supplements, are not entirely convincing. The Recall2 positive > control contrast shows no significant difference between conditions. Differences during the elaboration task are found in regions that might reflect generally higher effort (VLPFC, DLPFC), and not necessarily emotion regulation in particular.

Overall, the fMRI data from Experiment 4 are the least compelling part in an otherwise coherent series of experiments.

Other comments and questions:

- The statement that data are inconsistent with concurrent or competitive memory traces (see p.15 bottom) seems unfounded and needs to be explained. In my view, most data points reported in the manuscript (except maybe Exp. 3, Immediate Group) are consistent with the idea that participants create an alternative, more positive version of the original memory during the elaboration task, and are then more likely to retrieve this positive re-interpretation during the delayed recall.
- Were trials included in the behavioural and fMRI analyses where subjects indicated they could not think of a positive aspect? I might have missed this information in the methods section, if not it should be clarified.
- Experiment 1: Was the greater increase in positive feeling also correlated with more positive content and dissimilarity in other groups, or was it specific to Positive Group?
- Experiment 2: Statistics for the Group x Time ANOVAs missing.
- Experiment 4: The paired t-tests used to compare mean correlation coefficients across conditions need to be described in more detail, e.g., what was being permuted/held constant, and how were t-values computed from the empirical null distributions.
- Also regarding the fMRI analyses (Exp. 4), the authors should explain why a 5mm smoothing kernel was applied before subjecting data to the RSA analysis. RSA/multivariate pattern analyses are typically conducted on unsmoothed data.

Reply to Reviews for: NCOMMS-20-34250A-Z

We would like to thank the editor and the reviewers for their time and for providing us with very constructive and detailed feedback. We are pleased the reviewers found our experiments to be “well-crafted”, “novel, creative, meticulous,” and “timely,” found our replications across 4 experiments to be “impressive”, and believed this work could “make a solid contribution” and “advance our understanding of a technique for updating negative autobiographical memories in a relatively naturalistic way.” We appreciate the opportunity to revise and resubmit our manuscript and believe the manuscript has benefited tremendously as a result of addressing these points. Briefly, we have taken the reviewers’ excellent suggestions to add a broader discussion on memory updating and emotion regulation, to add methodological details to improve clarity, and to conduct additional analyses to rule out alternative interpretations of our findings. Below, we present a point-by-point response to each of the concerns raised by the reviewers. All revisions to manuscript text are included below for convenience but also depicted in the manuscript by red font. Thank you again for your time and helpful feedback.

REVIEWER COMMENTS

Reviewer #1 (Remarks to the Author):

In this paper, the authors outline four well-crafted experiments examining how finding positive meaning in negative autobiographical memories may utilize memory update mechanisms. Through their experiments, they show that memories that are manipulated only during a labile period are modified, and that changes persist for up to two months after modification.

I believe this paper is acceptable with minor revision. I list some areas of clarification below.

Comments:

1. There was a brief comment on the intersection of memory updating and emotion regulation in the discussion, but the experiments in this manuscript may benefit from being framed in the context of these existing studies. See Samide & Ritchey (2020): Reframing the Past: Role of Memory Processes in Emotion Regulation for a review, and Parikh, McGovern, & LaBar (2019): Spatial distancing reduces emotional arousal to reactivated memories for a recent empirical article.

Thank you so much for this suggestion. We have added a broader discussion of the intersection between memory updating and emotion regulation to our discussion section.

Manuscript excerpt (pg 17):

An innovative aspect of the present research is examining cognitive regulation as a tool for memory updating, beyond its known role in changing our current emotional state. Decades of research highlight cognitive regulation as being highly effective in reducing negative feelings

Reply to Reviews for: NCOMMS-20-34250A-Z

associated with an adverse event by changing how we think about it^{4,18,30}. Longitudinal investigations started to hint at the idea that regulation could have a lasting impact on emotion and memory, for instance, by showing that repeated regulation training can lead to a persistent reduction in negative reactivity to the same stimuli across time^{31,32}. However, these studies primarily used non-naturalistic stimuli (e.g., IAPS images or video depictions) that are less self-relevant and may rely less on memory systems than when thinking about one's own historical past³³. Studies examining the interaction between cognitive regulation and autobiographical memory found similar reductions in negative feelings that mirror the regulation of ongoing experiences, but only focused on immediate or short term effects (up to 30min later)³⁴⁻³⁷. That finding positive meaning in past negative events can change both how we feel and what we remember in the future provides evidence of the multifaceted role and long-lasting impact of cognitive regulation strategies on psychological wellbeing.

2. In Experiment 1, you report a correlation between positive feeling change and content dissimilarity from recall 1 to recall 2. Is this change fully explained by new positive content incorporated into the memory descriptions? If not, can you discuss the ethical and therapeutic implications of altering the content of recalled memories?

Yes, our analysis of memory content in Exp 2 showed that future retrieval included a combination of the original details, positive details from the elaboration period, and some new related positive details of the event. A key goal when designing these studies was to examine a regulation strategy that naturally occurs in everyday life. Positive meaning finding is not only explicitly taught in therapy, but it's also one of the most efficacious and widely used strategies in everyday life. This is true even outside of therapeutic contexts—whether one is thinking about a negative event on their own or talking to a friend about it. Together, this made it an excellent choice for balancing ecological validity with ethical concerns. Since participants are never explicitly asked to forget aspects of their memory or change their memories with artificial information, the ethical and therapeutic implications ought to mirror what naturally occurs in everyday life or in a therapy session when individuals learn to reframe a past negative event in a more positive light. We hope that this research can start to untangle what is happening to our memory over time, little by little, when we use this strategy. It could also help us to identify when and for what reason some individuals have difficulty updating their memories in a positive way, which hopefully could inform future treatment. We now highlight this in the discussion section.

Manuscript excerpt (pg 19-20):

Finding ways to lessen the deleterious impact of negative autobiographical memories has long captured the attention of researchers and is a prominent objective in therapeutic contexts⁴². The present research highlighted one such strategy. Not only do people already use positive meaning finding in their daily lives³⁸, but it also does not ask individuals to forget aspects of their memory or change their memory with artificial information, giving it high ecological validity.

Reply to Reviews for: NCOMMS-20-34250A-Z

3. Are you able to report full demographics of the participants in the study (e.g., ethnic/racial breakdowns?)

Yes, we have race/ethnicity data for Exp 1, 3, 4 and Supplemental Exp 5. We did not collect race/ethnicity data for Exp 2. We have added these demographics to the methods section for each study.

In Exp 1 methods:

The race of this sample was 16.7% Asian, 22.5% Black, 8.8% Middle Eastern or North African, 2.0% Pacific Islander, 32.4% White, 3.9% Other, 2.0% more than one race, and 11.8% did not report their race. Their ethnicity was 23.5% Hispanic, 75.5% non-Hispanic, and 1.0% Other.

In Exp 3 methods:

The race of this sample was 26.4% Asian, 31.9% Black, 5.6% Middle Eastern or North African, 18.1% White, 4.2% Other, and 13.9% did not report their race. Their ethnicity was 26.4% Hispanic and 73.6% non-Hispanic.

In Exp 4 methods:

The race of this sample was 37.5% Asian, 21.9% Black, 34.4% White, 3.1% Other, and 3.1% did not report their race. Their ethnicity was 18.8% Hispanic and 81.3% non-Hispanic.

4. Even within the restriction of negative, specific memories, memory quality and specific valence can vary greatly from individual to individual. For this reason, have you considered running the analyses as mixed models instead of ANOVAs with a fixed effect of participant? This analysis will also nicely mirror the set-up of the RSA, with each memory as an individual data entry.

Given the reviewer's suggestion, we ran our ANOVAs as mixed models with a fixed effect of participant. When we do this, it yields very similar output to our ANOVAs. For instance, in Exp 1, our group F-test statistic for the multi-level model comparing feeling change by group is $F_{3,98}=4.13$, $p=.009$, whereas the ANOVA yields $F_{3,98}=4.08$, $p=.009$. Since there is not a significant difference, we have chosen not to change our analyses to multi-level models. However, we now include a test for individual-level baseline differences in memory across conditions/groups in each study and control for such variables when deemed significantly different across conditions/groups.

5. Page 18: In Experiment 1, what were the reasons to base the power analyses off of a large effect? Was this purely a conservative choice?

Yes, we did make a conservative choice when selecting our initial effect size for the power analysis in Exp 1. Specifically, we predicted a medium effect size (Cohen's $d \sim .50$) for changes in emotion via positive meaning finding since similar paradigms vary in their effect sizes for this reappraisal strategy when regulating IAPS images or memories. We then based effect sizes for

Reply to Reviews for: NCOMMS-20-34250A-Z

the following Exp 2-4 on our results from Exp 1. We now describe these rationales in our methods section for each study.

In Exp 1 methods:

We conducted a power analysis using G*Power to determine sample size. We predicted a medium effect size as a conservative estimate based on similar paradigms⁴³, which yielded a target sample of 100 participants (25 per group; 80% power).

In Exp 2 methods:

Using G*Power, our target sample size was calculated to be 90 participants (45 per group) based on a power analysis expecting a small effect size (for detecting differences in written memory content as indicated in Experiment 1; 80% power).

In Exp 3 methods:

We used G*Power to determine sample size. We expected a large effect size (based on Experiment 1 results for emotion change across groups), which yielded a target sample size of 75 participants (25 per group; 80% power).

In Exp 4 methods:

Our target sample size was 35 participants based on prior fMRI studies using similar multivariate analyses and behavioral data from Experiment 3.

6. Page 21: What percentage of memories were removed due to your exclusionary criteria? I ask this because Experiments 1 and 2 already have very few memories per condition per person.

Participants reported 9.46 memories on average (out of 12 total). Therefore, about 21.2% of memories were excluded based on our exclusion criteria. We now report this information in the Exclusion criteria section of the methods (pg 23).

7. Similarly, can you reiterate how many participants were excluded in Experiment 1 due to the exclusionary criteria on page 21 (within the exclusion criteria section)? Though these numbers are reported in the participants section, listing them side by side with the exclusionary criteria would make it easier to quickly perceive whether participants were generally compliant.

As requested, we have integrated these details into the exclusion criteria section. Since our exclusion criteria section is for all studies in general, not just Exp 1, we also added a reference to our Supplement that contains a table with the specific counts by each exclusion type for Exp 1-3 (see pg 23 of methods & Supplementary Table 1).

8. In Experiment 3, you collected a date for each memory to ensure each memory had a specific time and place. Are these dates precise enough to use as a measure of time since event? With evidence of episodic memories becoming more semantic/crystallized

Reply to Reviews for: NCOMMS-20-34250A-Z

over time, one interesting addition to this work would be to see if more recent memories are more affected by the positivity manipulation than older memories.

Unfortunately, the dates are not precise enough to meaningfully interpret positivity differences by age. Because the age of memory was not a primary hypothesis, we did not collect precise dates, but rather time ranges to make sure that memories were similar in age across conditions (e.g., within the past 1-month vs. 5 years ago). However, we agree with the reviewer that this would be a very important and interesting question for a future inquiry.

9. Page 28: Could you clarify which preprocessing steps were completed in SPM12? It looks as though the first paragraph describes SPM12 preprocessing steps while the additional preprocessing (e.g., motion spikes) were in FSL, but the former is not explicitly stated.

We apologize for the confusion. The reviewer is correct that the initial preprocessing steps were carried out in SPM12, and then the motion correction was done using tools from FSL. We have revised this section of our methods to clarify this (pg 30).

10. Page 29: The whole brain fMRI analyses are set up here, but there is no explicit mention in the methods that the full analyses are in the Supplement. Furthermore, could you include a list of contrasts of interest from the whole brain GLM in the main text for additional clarity?

We now highlight some findings from our whole brain analyses in the main text and refer to the full analysis in the Supplement. As requested, we have also added a full list of all contrasts we report in Supplement to the main text (see below) and also to the methods section.

Manuscript excerpt (pg 15):

In addition to our key analysis, we also performed whole-brain analyses contrasting positive and control trials during each task. During the Elaboration task, positive meaning finding (relative to natural recollection) yielded activity consistent with prior studies examining positive reappraisal in particular (ventral striatum, caudate and vmPFC)^{16,17} and cognitive reinterpretation more generally (VLPFC, DLPFC and DMPFC; Supplementary Fig. 2)^{18,19}. Interestingly, we observed similar activation during Recall 2 for positive relative to control trials when tracking increases in positivity on a trial-by-trial basis, suggesting that updated memories may re-engage some of the same corticostriatal circuitry that was previously engaged during positive elaboration. (See Supplementary Information for whole-brain analyses of positive > control trials during the Elaboration task, Recall 1, Recall 2, Recall 2 parametrically modulated by positive feeling, parametric regression of neural activity during positive elaboration that tracks pattern dissimilarity across retrievals, and correlations between activity during Elaboration and future feeling change).

Reply to Reviews for: NCOMMS-20-34250A-Z

Reviewer #2 (Remarks to the Author):

Overall, I thought that this was a novel, creative, and meticulous paper that reported on some interesting findings. I definitely think that this paper should be published and am certain that it will make a solid contribution to the literature. I just had a few follow-up questions and suggestions, hopefully to improve the paper.

1. I thought that one of the most novel parts of the paper was the integration of a reconsolidation account. However, this seemed to be a bit buried in the paper. Thus, the introduction moves very quickly through the reconsolidation literature and the motivation for these studies. Only one sentence is spent describing the reconsolidation literature (beginning of second paragraph), which is central to the hypotheses presented. I would suggest adding a bit more to the introduction to review the relevant points from the reconsolidation literature. I would also suggest mentioning reconsolidation in the title.

We appreciate the reviewer's positive evaluation of our paper. Although we certainly agree with the reviewer that the reconsolidation study is one of the most novel parts of the paper, we have chosen to refrain from mentioning reconsolidation in the title or framing this work specifically around the reconsolidation literature, since only 1 of the 4 studies explicitly examine reconsolidation as a potential mechanism for memory change. However, we agree with the reviewer that the reconsolidation literature is important for understanding Exp 3 and this work more generally. Thus, we now emphasize the reconsolidation literature in our results and discussion sections.

Manuscript excerpt (pg 16-17):

Here, introducing new relevant information (i.e., positive reinterpretation) after reactivating an existing memory led to an integration of the positive reinterpretation into the negative memory trace, thus modifying future recollections in a beneficial way. This conceptualization differs from alternative mechanisms that could also lead to future memory change, such as concurrent or competing retrieval of dual memories²², which was not supported by our data (see Exp 3). That is, updates were only observable after a previous reminder during the reconsolidation window (manipulation occurring ~10 min to 6h after reactivation) and a memory test after a sufficient delay (24h later). Importantly, we assume there to be similar mechanisms across studies given that all other paradigms mirrored a version of the *Delayed-test* group in Exp 3. Similar evidence of updating has been observed across various memory domains. For instance, in associative memory, extinction after reactivating conditioned fear memories can reduce physiological arousal at future retrieval in humans²³ and freezing behavior in rodents²⁰ through a reconsolidation process

2. Along the lines of reconsolidation, can the authors clarify the time passed between the Recall (reminder) and Elaboration (intervention) tasks for these studies, and how this speaks to the reconsolidation literature (whether the reconsolidation window is "open" during elaboration)? My understanding is that at least 10 minutes should pass between

Reply to Reviews for: NCOMMS-20-34250A-Z

the reminder cue and memory-updating intervention: Monfils M. H., Cowansage K. K., Klann E. & LeDoux J. E. Extinction-reconsolidation boundaries: key to persistent attenuation of fear memories. *Science* 324, 951–955 (2009). It would be helpful to have a bit more clarity on this.

In our reconsolidation study (Exp 3), there is always a gap of at least 10 min between reactivation (Recall 1) and the manipulation (Elaboration) consistent with the reconsolidation literature. We have provided clarity on this when describing the experimental paradigm in Exp 3.

Manuscript excerpt (pg 10):

Importantly, since the updating window is proposed to begin at least 10 min (and up to 6h) after reactivation¹⁴, the positive manipulation (Elaboration) occurs 10 min following reactivation (Recall 1) in the two groups where updating is meant to occur within the updating window (*Immediate-Test* and *Delayed-Test* groups).

3. Did the authors differentiate between negative feeling change and positive feeling change? While it may be the case that an increase in positive feelings corresponds to a decrease in negative feelings, it is also possible that there is no change in negative feelings associated with the autobiographical memory alongside a significant change in positive feelings. This seems like an interesting and important consideration in explaining how these memories are updated, and is relevant in terms of situating this work in the reconsolidation literature since it seems like a vast majority (but not all) of reconsolidation paradigms focused on attenuating the negative affect associated with certain stimuli, whereas this paper focuses on increasing positive affect associated with particular stimuli. It looks like the scale used to assess participants' feelings ranges from negative to neutral to positive, so could the change in positive feelings described in the results be due to a decrease in negative affect as opposed to an increase in positive affect? It would help if you could methodologically clarify how this scale was used, and also discuss the theoretical implications associated with this focus on increasing positive affect versus decreasing negative affect associated with negative autobiographical memories and how this relates to other reconsolidation work.

We did not differentiate between negative and positive feeling change. Our scale was on a continuum from very negative (-5) to very positive (+5). Therefore, we focused primarily on the change in emotion across this scale. Since our goal was to examine feeling change from negative to more positive across time as proof of concept with respect to memory change, we did not focus on more nuanced questions regarding decreases in negativity vs. increases in positivity vs. a combination of the two. It is also unclear to what extent individuals are able to accurately rate whether they are experiencing only a decrease in negativity vs. increase in positivity vs. both, and this will likely differ across individuals as well. However, studies examining positive meaning finding have shown that this strategy leads to both an increase in positivity as well as a decrease in negativity, unlike other regulation strategies (e.g., suppression, distancing). Therefore, our assumption would be that both are likely occurring. We now mention this in our discussion section.

Reply to Reviews for: NCOMMS-20-34250A-Z

Manuscript excerpt (pg 19):

We used a rating scale across a continuum from negative to positive feeling, but future paradigms may want to tease apart changes in positive emotion and negative emotion separately. This would be particularly important for testing within a reconsolidation paradigm as such studies primarily targeted negative emotion.

4. While I very much appreciated the ‘dissimilarity’ analyses, I also found myself wondering if subjects in the positive reinterpretation condition showed general increases in ventral striatum and VMPFC activity during scan #2 relative to scan #1 (in the recall period). I later saw that these analyses were included in the supplementary materials. I would suggest considering moving these results to the main text, as my guess is that other readers will have these same questions.

We appreciate this suggestion which was also echoed by other reviewers. We now highlight some of the key findings from our univariate whole-brain analyses in the main text, as requested. Since these additional analyses were not key hypotheses in our fMRI experiment and due to space limitations, we did not move all univariate analyses to the main text. However, we do provide a full list of additional contrasts (reported in the Supplement) so that readers will know the full extent of analyses performed and can reference the Supplement for those results.

Manuscript excerpt (pg 15):

In addition to our key analysis, we also performed whole-brain analyses contrasting positive and control trials during each task. During the Elaboration task, positive meaning finding (relative to natural recollection) yielded activity consistent with prior studies examining positive reappraisal in particular (ventral striatum, caudate and vmPFC)^{16,17} and cognitive reinterpretation more generally (VLPFC, DLPFC and DMPFC; Supplementary Fig. 2)^{18,19}. Interestingly, we observed similar activation during Recall 2 for positive relative to control trials when tracking increases in positivity on a trial-by-trial basis, suggesting that updated memories may re-engage some of the same corticostriatal circuitry that was previously engaged during positive elaboration. (See Supplementary Information for whole-brain analyses of positive > control trials during the Elaboration task, Recall 1, Recall 2, Recall 2 parametrically modulated by positive feeling, parametric regression of neural activity during positive elaboration that tracks pattern dissimilarity across retrievals, and correlations between activity during Elaboration and future feeling change).

Some additional minor issues:

5. Figure 1a: The last panel in “Memory Recall 2” says: “Same memory as last session?” Is this question asked to participants, or determined by coders? This figure makes it seem like the former (which seems odd to ask), but the paper description makes it seem like the latter.

Yes, we do ask participants to report whether they retrieved the same memory as the last session (meaning, was this the same event triggered by this cue, or were you unable to think of that event?). This was simply to ensure that changes in emotion/content are not due to thinking

Reply to Reviews for: NCOMMS-20-34250A-Z

of an entirely different past event at future recall. Although this did not happen frequently, this is helpful for cases in which event cues were not very specific or descriptive and participants may have forgotten which specific memory it corresponded to. This was particularly helpful in the mental recall paradigms (Exp 3 & 4) since there was no written content for coders.

6. Exclusions for Experiments 1-3 by Group: The third category (remembered <50% of memories) is confusing. Can this be explained differently?

Yes, thank you for this suggestion. We have revised this to improve clarity. It now reads: “had fewer than 50% of memories that met criteria*” and a footnote that reads: “*This was due to participants not following directions or reporting it was not the same memory across retrievals.” See Supplementary Table 1.

7. Supplemental Methods & Results for Experiment 1 – Positivity of Content Rating: The description suggests that the rating is based on negativity of the description, whereas the scale is based on similarity of the description. Can you clarify this discrepancy? Small typo on this scale: “Very Dissimilarity”

We have fixed this typo.

8. Can you clarify on page 4 of the supplement whether Recall 1 and Recall 2 are both from the “Write description” part of the tasks (and not the “Write elaboration” part?

Recall 1 and 2 are only from the “Write Description” part of the tasks on shown in Figure 1 panel a. The “Write Elaboration” part is only referred to as Elaboration, rather than one of the retrieval sessions.

9. Experiment 2 supplement: Can you include the prompt for the natural recollection group (i.e., control condition)?

Another reviewer asked us to move the Exp 2 analyses to the main text, and therefore it no longer appears here. The prompt for natural recollection appears in the main text.

10. It might help to add some information about the reconsolidation window to Figure 3A, and to clarify that Recall 1 is serving as the “reminder cue” before the window is opened, Elaboration is introducing a manipulation (hopefully during the reconsolidation window), and Recall 2 is testing whether the memory was updated. As is, it takes some effort to puzzle through the conditions introduced in the figure.

Thank you for this suggestion. As requested, we have added a description to the figure caption for Fig 3A stating that Recall 1 serves as reactivation, Elaboration is the manipulation, and Recall 2 is the future memory test.

11. Can you provide the Autobiographical Memory Questionnaire you used in your

Reply to Reviews for: NCOMMS-20-34250A-Z

supplemental materials so that readers can get a complete sense of the negative event cues?

We have added a list of the negative event cues from our AMQ to our Supplement.

12. Thank you for making your data available on OSF, but it does not appear to be publicly available. Can you share access with reviewers? Can you additionally share the code used to run your analyses in your repository?

Yes, here is a link to the behavioral data and code for the 4 experiments reported in the paper:

https://osf.io/jtqfk/?view_only=5565e09d0dbf43b59bd98f4af58ba55b

We plan to make the behavioral data and code publicly available at the time of publication.

Reply to Reviews for: NCOMMS-20-34250A-Z

Reviewer #3 (Remarks to the Author):

This study used a series of behavioral experiments and an fMRI experiment to test whether focusing on positive aspects of a past negative event can adaptively update the negative memory with more positive feelings and content, ultimately changing how the memory is represented in the brain. The authors report that a positive elaboration manipulation elicited greater increases in positive feelings than negative, neutral and distraction manipulations after 1 week (behavioral Exp 1). The positive elaboration manipulation also yielded greater increases in positive feelings, positive content and memory dissimilarity after 1-week compared to a natural recollection condition, and the effect persisted for 2-months (behavioral Exp 2). In behavioral Exp 3, they report findings consistent with a reconsolidation-based account of the memory updating mechanism. Using fMRI (Exp 4), they examine whether the neural representation of memories becomes more dissimilar across recalls following positive elaboration. Although the positive elaboration and control condition do not yield overall differences in representational dissimilarity, the authors do find evidence that changes in positive emotion are correlated with neural dissimilarity across memories in the hippocampus and ventral striatum.

I am impressed with the authors' commitment to replicating their results across multiple experiments. These findings advance our understanding of a technique for updating negative autobiographical memories in a relatively naturalistic way and provide insight into the underlying mechanism. The methods appear to be sound. I have a few questions and suggestions:

1. It might be informative to specifically test if the relationship between neural dissimilarity and change in feeling rating is indeed memory-specific. For example, you could shuffle the pair assignments of neural dissimilarity and feeling rating and recompute the spearman correlation X number of times in each participant (within condition). If the relationship is indeed item-specific, then the observed value should be larger than the permuted distribution. For statistical analysis, you can compute the z-score of the true value relative to the mean and SD of the permuted distribution in each subject. Then examine the reliability of these z-scores across participants with a one-sample t test against zero.

This is a very interesting idea. As requested, we conducted this analysis following the steps suggested by the reviewer. We shuffled the pair assignments of neural dissimilarity and feeling change and recomputed the spearman correlation (10,000 permutations) to create a permuted distribution. We did this for each ROI (hippocampus, NAcc) within each condition and participant separately. We found that the observed value was significantly greater than the permuted distribution, implying that this relationship is indeed memory specific, in the hippocampus but not the NAcc. We added this analysis to our Supplement.

Supplement excerpt (pg 12-13):

Reply to Reviews for: NCOMMS-20-34250A-Z

RSA: Memory specificity of Recall 1 - Recall 2 pattern change relationship with feeling ratings

We performed an additional analysis to test whether the relationship between neural dissimilarity and feeling change was memory-specific. We first created permuted distributions of this relationship by shuffling the pair assignments (10,000 permutations) of neural dissimilarity and feeling change and then computing our spearman rho correlation for the two ROIs (hippocampus, NAcc) within each condition and participant separately. We then compared the mean of our observed values with the mean of the permuted distribution across participants for each ROI in the positive condition. Our observed correlation was significantly greater than the permuted distribution in the hippocampus ($t_{31} = 2.83$, $p = .008$), suggesting that the relationship between greater neural dissimilarity and greater increases in positivity across time is indeed memory-specific. Although it was in the expected direction, we did not observe this in the NAcc ($t_{31} = 1.53$, $p = .136$).

2. In Exp 4, is there a way to test whether the recall 2 memories do indeed integrate more “positivity” in their neural representation, which would account for the increased dissimilarity from recall 1 to recall 2 (specifically in the positive relative to control condition)? This might not be possible in this particular design, but one idea is maybe you could use the fMRI data from the elaboration task to estimate an average “template” of positive emotion during recollection (perhaps by simply averaging across positive elaboration trials or contrasting positive > natural), then regressing out this template from recall 2 patterns (both positive and control trials) and re-running the neural dissimilarity/emotion change RDM correlations.

We appreciate the reviewer’s suggestion. We agree with the reviewer’s point that it may not be possible to conduct this analysis with our particular paradigm. Our paradigm was designed primarily to examine neural pattern dissimilarity across retrievals, but not to identify a positivity template during elaboration. Since the positive trials in the elaboration task still include details of the negative memory, it would be challenging to identify a clean ‘positivity’ template without it highly overlapping with neural activity associated with memory recollection. Additionally, the positivity template may also differ from memory to memory, making it unclear how well it could answer the question asked by the reviewer.

However, we were interested in and explored a similar question that we reported in our Supplement. Specifically, we tested whether activity during the elaboration period (e.g., striatum or prefrontal cortex) predicted future memory change, both in behavior (feeling change) and neural pattern change. We found that a greater increase in positivity across retrievals (Recall2 – Recall1) was associated with greater activity in the DLPFC ($r_{31} = .450$, $p = .009$) and VLPFC ($r_{31} = .354$, $p = .043$) during positive elaboration, which may reflect greater regulation success (Wager, Davidson, Hughes, Lindquist, & Ochsner, 2008). We also conducted a whole brain parametric regression analysis for memories in the Elaboration task weighted by their neural dissimilarity across retrievals (in the hippocampus and striatum separately), as well as targeted analyses within prefrontal and reward-related ROIs, but found no significant relationships. These null results could be due to a lack of power or an incompatibility with the task design,

Reply to Reviews for: NCOMMS-20-34250A-Z

among other things. Nevertheless, it will be important for future research to design studies to examine this question.

Minor points:

3. (page 6) Please remove the word “trending” when describing a statistical test where $p > 0.1$.

We have removed this and instead stated it did not reach significance but is in the expected direction.

Manuscript excerpt (pg 5):

The one-way ANOVA for change in positive content (Recall2–Recall1) was in the expected direction but did not reach significance ($F_{3,98}=2.02$, $p=.116$, $\eta^2=0.058$) and for content dissimilarity was non-significant ($F_{3,98}=1.53$, $p=.326$, $\eta^2=0.035$).

4. (page 8) I’m noticing the df (241) reported for the correlations of future-oriented elaborations with positive feeling change and positive content change. Is this correct? Do these correlations include multiple data points per subject? If so, you should eliminate the dependencies in the correlation.

We have replaced these statistics with correlations for subject-level data. The results remain unchanged.

Manuscript excerpt (pg 7):

Importantly, within the *Positive* group, future-oriented elaborations were not correlated with positive feeling change ($r_{24}=-0.11$, $p=.569$) or positive content change ($r_{24}=0.028$, $p=.891$), suggesting that a focus on future outcomes did not drive our observed changes in memory across time.

5. (page 8) Please report the statistics for the time x group ANOVA main effects and interaction in the main text (currently they are in the supplement). Also, the t-tests reported in the main text and supplement are redundant. I also noticed that some of the Experiment 3 and 4 results from the main text are repeated in the supplement.

We have added the ANOVA main effects and interactions to the main text from the supplement. We intentionally included some of the same results and descriptions from the main text when describing the supplemental Exp 4 results only to provide continuity and additional clarity for readers, but we have removed any areas that seemed too redundant with the main text.

Manuscript excerpt (pg 7-8):

We examined changes in feeling ratings and memory content (positivity, dissimilarity) via Time (1-week, 2-months) by group (*Positive*, *Control*) ANOVAs, which yielded significant main effects of time ($F_{1,178} = 11.82$, $p < .001$, $\eta^2 = .023$; $F_{1,178} = 7.28$, $p = .008$, $\eta^2 = 0.037$; $F_{1,178} = 21.31$, $p <$

Reply to Reviews for: NCOMMS-20-34250A-Z

.001, $\eta^2 = 0.098$) and group ($F_{1,178} = 4.43$, $p = .037$, $\eta^2 = .061$; $F_{1,178} = 9.10$, $p = .003$, $\eta^2 = 0.047$; $F_{1,178} = 19.83$, $p < .001$, $\eta^2 = 0.091$), but no interactions ($p = .824$; $p = .840$; $p = .644$).

6. (page 9) “Participants reported 4.74 details per memory on average (SD=1.36 range= 2.3-9.0), with no difference between groups” Is this during Recall 1 or all recalls together?

This is during Recall 1. We now clarify this in the manuscript text (pg 9).

7. (page 10) How are the “new related positive detail” percentages calculated (percentages of what)? Are new related positive details at 2 months new relative to session 1 or new relative to session 2? Are you also able to report the percentage of new negative details? The authors report that the groups did not differ in the number of new negative details, but I am also curious about how the number of new negative details compares to the number of new positive details.

The new related positive and negative detail percentages were calculated as the number of new details divided by the number of total details during retrieval. That is, the percentage of details in the future retrieval that included ‘new’ details rather than old ones. We have added a description of how this percentage was calculated when describing these results in the manuscript. Additionally, we now report the number of new negative detail percentages for each group in the manuscript text.

Manuscript excerpt (pg 9):

Their future recollections also included 12.2% and 14.1% of new related positive details at 1-week and 2-months, which was significantly greater than the *Control* group (1-week: 8.1%, $t_{89}=2.63$, $p=.01$, $d=0.55$; 2-months: 10.2%, $t_{89}=2.29$, $p=.025$, $d=0.48$; percentage of new details corresponds to the number of new details divided by the total number of details during retrieval). Groups did not differ in the number of new negative details that were present at 1-week ($M_{\text{Positive}}=10.9\%$, $M_{\text{Control}}=8.3\%$; $t_{89}=1.74$, $p=.085$, $d=0.37$) or 2-months ($M_{\text{Positive}}=10.7\%$, $M_{\text{Control}}=8.8\%$; $t_{89}=1.42$, $p=.16$, $d=0.30$).

8. (page 10) “Together, these findings suggest that positive meaning finding leads to future recollections with components of both the initial recollection and the positive elaboration, lending support to a memory updating account.” Do the results also indicate that they produce new positive details too (that weren’t part of the initial recall or positive elaboration)?

Yes, the positive group did show a slightly greater increase in new related positive details than the control group (1-week: 12.2% vs. 8.1%; 2-months: 14.1% vs. 10.2%), whereas both groups had similar increases in new related negative details that did not significantly differ from each other (1-week: 10.9% vs. 8.3%; 2-months: 10.7% vs. 8.8%). The fact that both groups—even the control group—retrieved new related positive and negative details in the future mirrors the reconstructive nature of memory. However, since the Positive group additionally incorporated

Reply to Reviews for: NCOMMS-20-34250A-Z

about ~10% of their positive elaboration in the memory trace (relative to 0% in the Control group—since they did not positively elaborate), we might expect the Positive group to recall more new positive details related to the event than the Control group in the future as well. Intuitively, this could be due to an extension of the incorporated positive content (from the elaboration) at future recall.

9. (page 11) Please report the condition x group interaction.

We have added the non-significant condition x group interaction to our analysis for Exp 3 (pg 12).

10. (Page 13) “Our key hypothesis was that positively reinterpreted memories (i.e., positive trials) would have greater dissimilarity in their neural activation patterns across retrievals in comparison to naturally recalled memories (i.e., control trials).” My understanding is that this isn’t exactly the key hypothesis that is then tested and reported in the subsequent part of the paragraph. You aren’t testing for the difference in neural dissimilarity between positive trials and control trials, but rather the difference in the relationship between neural dissimilarity and emotion change between groups.

We agree with the reviewer that our stated hypothesis was too broad. We have revised our statement to clarify that we’re testing for greater dissimilarity linked to emotion change across groups.

Manuscript excerpt (pg 13):

Our key hypothesis was that positively reinterpreted memories (i.e., positive trials) would have greater *dissimilarity* in their neural activation patterns across retrievals as a function of increasing positivity in comparison to naturally recalled memories (i.e., control trials).

11. (methods - Exp 1) During session 2, were subjects specifically instructed to recall the same memory for each cue (e.g. family vacation) as they did during session 1? Or were they just shown the cue and they could choose to recall a different memory?

They were asked to recall the same memories as in session 1. We also asked them to verify that it was the same memory as the last session, and we only included memories that met this criterion. We have clarified this in the methods section for Exp 1.

Manuscript excerpt (pg 22):

They were also asked whether they recalled the same memory as the last session, so we could ensure that memory change was not due to recalling a different memory in the future.

12. Supplementary info page 3 - The scale listed under positivity content rating is not correct. For the dissimilarity rating scale, I believe it should read “Very Dissimilar” not “Very Dissimilarity”

Reply to Reviews for: NCOMMS-20-34250A-Z

We have fixed this typo.

13. Fig S1 - when you say collapsing across 1-week and 2-month retrievals, does this mean that there are two data points per subject included in the correlation? I would prefer seeing this plot and correlation coefficient separated for 1-week and 2-month retrievals.

We have updated Supplementary Fig 1 and the description of this analysis in the Supplement to include separate correlations with 1-week and 2-month retrieval periods.

14. There are some typos in second to last sentence on page 16 of supplement.

We have fixed these typos.

Reviewer #4 (Remarks to the Author):

Review of “Finding positive meaning in past negative events adaptively updates memory”

This paper offers evidence to the effect that revisiting memories of past negative experiences within the content of finding positive meaning to them, updates the underlying memory representation in adaptive ways. It is a timely paper, and the result could potentially help to illuminate the cognitive and neural mechanisms underlying behavior change in therapeutic contexts. That being said, I have a number of concerns that prevent me from recommending this manuscript for publication in its current form. I elaborate on these concerns, below.

1) p. 5. I found the instructions to the “meaning making” manipulation a bit open ended. It seems to me that “describe something you learned”, “think about something positive that occurred as a result of this”, and “how the event is meaningful to you”, are rather different elaboration strategies that likely bring about distinct cognitive processes and emotions. Thinking about something that occurred as a result of a past event involves causal reasoning and likely the recollection of a subsequent episodic autobiographical memory, and we do not know if that memory is positive or negative. By contrast, thinking about how the event is meaningful to oneself may not require causal reasoning, or the retrieval of a subsequent memory, but rather abstraction or some other kind of conceptual operation. My concern is that there are a number of different kinds of processes involved here, and it is not clear which one in particular may be doing the work. Notice, incidentally, that the instructions for the other two conditions are less open ended—at least from the description in the paper.

The strategy of positive meaning finding is subjective and meant to inspire thinking that is unique to the specific event or situation. But this is not unlike most—if not all—other cognitive regulation strategies, including the negative-focus strategy we test in Exp 1. Therefore, we don't consider that elaborating in a positive way vs. negative way would necessarily differ in its degree of open-endedness. That is, when thinking about why something is negative (which is akin to rumination), this also could involve causal reasoning (as to why) and/or a recollection of a subsequent episodic autobiographical memory (e.g., initial memory = bad grade on exam; subsequent memory = failed out of school). Mirroring this thinking, in a previous exploratory analysis that we already report in the main text, we coded elaborations into those that involved thinking about a subsequent event (e.g., future-oriented) vs. focusing on the event itself. We found no difference in the degree to which participants used future-oriented vs. non-future-oriented elaborations between the positive and negative groups. Within the positive group, there was also no difference in outcome (e.g., emotion change) based on whether the positive elaboration focused on a subsequent memory (future-oriented) or finding meaning in the event itself. Based on these data, it would suggest that there isn't a specific route to positive meaning finding that is 'doing the work' but rather whether or not this strategy was used. To address the

Reply to Reviews for: NCOMMS-20-34250A-Z

reviewer's concern, we have revised the instructions of the negative group to make it clear that it too can involve causal reasoning and future-oriented thought, much like the positive group.

Manuscript excerpt (pg 4):

...(Negative group; n=25; e.g., describe what makes this memory negative or something negative that occurred because of this)...

2) p. 5. Last two lines. What is the rationale behind this hypothesis?

The end of pg 5 outlines our key hypothesis for this research, which we previously motivated in the introduction section. We predict that positive meaning finding will lead to greater changes in emotion and memory content compared to other strategies such as: focusing on negative aspects (akin to analyzing/problem solving or rumination), focusing on neutral aspects (akin to distancing), or by using distraction. First, this strategy is the only one amongst these that leads to an increase in positivity as well as a decrease in negativity, leading to a greater overall emotion change (Dore et al., 2016). Second, this strategy may be more likely to trigger a prediction error given its greater divergence from the negative affect of the retrieved memory than the other strategies, which may be necessary for memory updating (Elsev, Van Ast, & Kindt, 2018; Sevenster, Beckers, & Kindt, 2013). We describe these differences in the discussion section, but now also re-emphasize the link between our strategy and positive psychological wellbeing (e.g., lower depression, greater positive emotionality, and faster stress recovery) when describing our hypothesis in Exp 1.

Manuscript excerpt (pg 4-5):

We hypothesized that only the *Positive* group would show enhanced positive feelings and the greatest change in memory content at future retrieval, given the link between positive-emotion focused coping and fewer depressive symptoms, more positive emotionality⁴, and faster recovery from stress⁵.

3) p. 5. (Also, p. 21, in methods). If I understand correctly the exclusions reported in the Supplemental Information, your ns per group are smaller than what reported in page 6. Specifically, the n for the positive group is 19 (26 – 7), for the negative group is 17 (25 – 8), for the neutral group is 17 (25 – 8) and for the distraction group is 20 (26 – 6). Is this correct? Additionally, you mentioned that you excluded those participants that remembered 50% or less of the same memories in S2 relative to S1. I had to dig into the supplemental information to learn that, on average, it looks like participants contributed 9.46 memories, although we don't know how many per group—all we know is that they did not differ statistically from one another. I think it may be important to frontload some of that information. I started reading experiment 1 thinking that each participant contributed on average 300 observations (25 x 12), but it turns out that they are effectively contributing a bit over half of that (~170). Alternatively, I may be wrong, and you are reporting ns *after* exclusions in p.5. I apologize if I misunderstood.

Reply to Reviews for: NCOMMS-20-34250A-Z

We apologize for the confusion. As you noted in your last sentence, we are in fact reporting the ns per group *after* exclusions on p. 5 as is typically done in a short format journal article. In the methods (p. 21) we report the total n and ns per group both before and after exclusion. Here is an excerpt from the methods section describing this: **“Participants in this 2-day study included 131 healthy young adults** who were randomly assigned to four experimental groups (Negative=33, Positive=33, Neutral=33, Distraction=32)... The **final sample included 102 participants** (35 men; Mean age= 20.3; SD= 2.9) across the Negative (N=25; 9 men), Positive (N=26; 8 men), Neutral (N=25; 8 men), and Distraction groups (N=26; 10 men).” The final 102 participants and ns per group matches what is described in the results section. We appreciate the reviewer raising this and to address this confusion, we have revised the participants section in Exp 1 to reference our Supplementary Table 1 that breaks down each exclusion type by group.

4) p. 8. Regarding experiment 2, were there any differences in the initial emotion of the reported memories (i.e., in session 1)? I could not find this information in the SI, and I think it is important that there aren’t baseline differences in affect between the groups to fully compare the effects of the manipulation. I noticed that the researchers also collected vividness, significance, social closeness, and frequency of recall. Were there any differences between groups for these measures? May be important to consider them as potential covariates.

In Experiment 2, there were no group differences in baseline ratings of feeling ($t_{89} = -0.466$, $p = .642$), intensity ($t_{89} = 0.037$, $p = .971$), vividness ($t_{89} = 0.706$, $p = .482$), significance ($t_{89} = 1.09$, $p = .277$), social context ($t_{89} = -1.56$, $p = .123$), frequency of recall in daily life ($t_{89} = 0.774$, $p = .441$) or age of the memories ($t_{89} = -0.175$, $p = .861$), suggesting that the Positive and Control groups recalled memories of similar emotional quality during recall 1. Since there are no baseline differences, we did not include these variables as covariates in our analyses. Given the reviewer’s suggestion, we have added this information to the supplement (pg 5).

5) p. 11. Experiment 3. According to the authors, in the no-reminder group there is no reactivation of the memory prior to the manipulation. But how can participants engage in thinking about positive aspects of a memory if they are not recalling it? I seems to me that for the elaboration to work in the “no reminder group”, participants still need to reactivate the memory they are going to think about, in which case it looks as though what happens in that group shouldn’t be different from what happens in the “delayed test group”. Yet, the authors find differences. I guess it may be worth clarifying how participants can find positive meaning in a remembered event without first remembering the event in which they are supposed to find meaning.

It is important to note that remembering and reactivating a memory are not synonymous with each other. Each time we remember a memory, this doesn’t necessarily mean it has been reactivated. Therefore, we agree with the reviewer that one must retrieve some aspects of a memory in order to generate a reinterpretation of that event (as was done in the No-Reminder group during the positive manipulation). However, within the 20s elaboration time frame in Exp

Reply to Reviews for: NCOMMS-20-34250A-Z

3, participants are asked to 'focus on the positive aspects' of the negative event, which differs from being asked to solely retrieve a negative memory in full, as is done in the reactivation task (Recall 1) or the natural recall condition during elaboration. To be certain, we conducted a new analysis of written content from Exp 1 to quantify how much time participants spent recollecting the initial memory when generating a positive elaboration. In this re-analysis, we see that participants do in fact spend a majority of their time positively elaborating on the memory (73.8%, SD = 4.1) and this is significantly more time than is spent on rehashing the recall 1 details (26.2%, SD = 4.1; $t_{25} = 29.4$, $p < .001$). This suggests that generating a positive elaboration did not require recollecting the full details of a memory.

However, since participants in this re-analysis of Exp 1 followed the timeline of the Delayed-test group from Exp 3 (e.g., received a reminder prior to elaboration), we wanted to examine this in participants who wrote about their memories but followed the timeline of the No-Reminder group from Exp 3 (e.g., received a reminder 24h prior to elaboration, outside the updating window). Therefore, we also conducted a brief exploratory study for the purposes of this reply to ensure that participants are able to generate similar positive elaborations in absence of a recent reminder. We recruited 24 Rutgers University students to complete an online version of the study via Qualtrics surveys for research credit—a sample size consistent with the comparison sample (i.e., Positive group in Exp 1). Participants in this written No-Reminder sample also spent a majority of their time positively elaborating (76.8%, SD = 10.2) rather than recollecting the details provided in Recall 1 (23.2%, SD = 10.2; $t_{23} = 6.43$, $p < .001$), and this was no different than the written elaborations of individuals who received a recent reminder (i.e., Positive group in Exp 1; $t_{48} = 0.73$, $p = .467$). This demonstrates that even without a recent reminder, participants are able to generate similar positive elaborations to those who were recently reminded, and this did not require recollecting the full details of a memory to do so. This is consistent with prior research showing that the degree of subsequent memory modification depends on the degree of memory reactivation, e.g., low vs. high subjective sense of reliving an event (St Jacques, Olm, & Schacter, 2013). Therefore, although participants remembered some aspects of Recall 1 during elaboration, it is possible that this was not of sufficient quality of reactivation to leave the memory liable for modification.

6) Another point about Exp. 3. Participants gave ratings of feeling, emotional intensity, vividness, etc. Do memories in session 1 differ along any of these ratings? If so, it may be important to include as regressors in the analysis, as it could rule out potential confounds.

We conducted condition (positive, control) by group (*Delayed-Test, Immediate-Test, No-Reminder*) ANOVAs for each baseline rating of memory separately. We found no significant main effects of condition, group or interactions for baseline intensity ($F_{1,138} = 0.001$, $p = .980$; $F_{2,138} = 0.83$, $p = .436$; $F_{2,138} = 0.15$, $p = .864$), vividness ($F_{1,138} = 0.003$, $p = .959$; $F_{2,138} = 0.164$, $p = .849$; $F_{2,138} = 0.019$, $p = .982$), social context ($F_{1,138} = 0.001$, $p = .979$; $F_{2,138} = 0.21$, $p = .808$; $F_{2,138} = 0.043$, $p = .958$), or age of the memories ($F_{1,138} = 0.16$, $p = .691$; $F_{2,138} = 2.50$, $p = .086$; $F_{2,138} = 0.28$, $p = .756$). We did, however, observe a significant main effect of group for baseline feeling ratings ($F_{2,138} = 4.29$, $p = .016$), but no effect of condition ($F_{1,138} = 0.01$, $p = .923$) or

Reply to Reviews for: NCOMMS-20-34250A-Z

interaction ($F_{2,138} = 0.04$, $p = .962$). When exploring this further, we found that the *No-Reminder* group had significantly greater baseline feeling ratings than the *Immediate-test* group ($t_{47} = 2.81$, $p = .007$) while neither group differed from the *Delayed-test* group ($t_{45} = 1.30$, $p = .201$; $t_{46} = 1.52$, $p = .137$). Therefore, we controlled for baseline feeling ratings in our analyses. We include these analyses of baseline memory ratings in the Supplement. We also re-ran the analyses in the manuscript text controlling for baseline feeling ratings. Importantly, the outcome of the results do not change when including baseline feeling ratings in the analysis.

Manuscript excerpt (pg 11-12):

When checking for baseline memory differences, we found that the *No-Reminder* group had significantly greater baseline feeling ratings than the *Immediate-test* group ($t_{47}=2.81$, $p=.007$, $d=0.80$) while neither group differed from the *Delayed-test* group ($t_{45}=1.30$, $p=.201$, $d=0.38$; $t_{46}=1.52$, $p=.137$, $d=0.43$). Therefore, we controlled for baseline feeling ratings in our analyses (see Supplementary Information for analyses of all baseline ratings). A condition (positive, control) by group (*Delayed-Test*, *Immediate-Test*, *No-Reminder*) ANOVA for feeling change, controlling for baseline feeling ratings, revealed a significant main effect of condition ($F_{1,137}=5.32$, $p=.023$, $\eta^2=0.03$) and group ($F_{2,137}=3.43$, $p=.035$, $\eta^2=0.04$), but no interaction ($F_{1,137}=1.81$, $p=.181$, $\eta^2=0.02$). The *Delayed-Test* group had a significantly greater increase in positive emotion for positive relative to control trials as compared to the *Immediate-Test* ($t_{45}=2.44$, $p=.019$, $d=0.74$) and *No-Reminder* ($t_{44}=2.03$, $p=.048$, $d=0.66$) groups, who showed no such changes and also did not differ from each other ($t_{46} = 0.291$, $p = .772$, $d = .055$; Fig 3b).

7) p. Regarding experiment 4, it would be nice to know the rationale for picking those ROIs. I was surprised, for instance, not to see the amygdala included among the ROIs, given its role in emotion (it is included in the SI, but not in the main text, not sure why). I was also surprised not to see an effect for VMPFC, which has been strongly associated with self-referential processes. My guess is that the authors included it as an ROI, in part, because of this. What do you make of this null result (p. 14)?

We selected ROIs that we thought would mirror our behavioral findings in terms of changes in positivity and memory content. Therefore, we chose regions critical to processing positive affect and memory. The VS and VMPFC not only correlate with the experience of positive emotion but also are specifically linked to the positive meaning finding strategy that we use in our paradigm (Dore et al., 2016). We selected the hippocampus as well because of its prominent role in memory. We describe this rationale in both the introduction as well as in the results section for Exp 4. We now highlight this rationale again when describing the analysis in Exp 4.

We did not have a strong prediction about the amygdala because although individuals recalled negative emotional events, the amygdala has a strong role in processing salience, and it wasn't entirely clear in what way the salience (beyond negative valence) might change after using positive meaning finding. Since the amygdala contributes to memory processing and negative emotion, we included it as an exploratory RSA analysis in our supplement.

Manuscript excerpt (pg 13):

Reply to Reviews for: NCOMMS-20-34250A-Z

We tested this using representational similarity analysis (RSA) in *a priori* ROIs (hippocampus, VS, VMPFC). We selected the hippocampus given its role in memory retrieval¹² and the reward-related ROIs (VS and VMPFC) given their links to the subjective value and positivity of recollection¹³.

Minor comments:

8) p. 4. Second paragraph, line 2. What do you mean by “reopens”? Maybe “reactivates”?

“Reopens” is terminology sometimes used to refer to a memory being labile to modification. We have changed this to the preference of the reviewer to state “reactivates.”

9) p. 6. Third paragraph, line 3. I don’t think that the result of that first ANOVA should be taken as “trending”.

We have removed the word ‘trending’ from this result.

Reviewer #5 (Remarks to the Author):

This manuscript reports 4 experiments (3 behavioural, 1 behavioural + fMRI) examining how positive re-evaluation changes the content of originally negative autobiographical memories. All experiments show robust effects on behaviour, such that positive reinterpretation immediately following memory reactivation leads to a lasting increase in positive emotional content. Overall, this is a well-conducted set of experiments with highly coherent results.

The editors specifically asked me to comment on the RSA approach in Experiment 4 (fMRI). Here, the authors test the key hypothesis that positive re-evaluation causes a change in a memory's neural representation, such that the neural patterns become more dissimilar when comparing the original Recall1 with the delayed Recall2. To test this hypothesis, participants are scanned twice, 24h apart, while they recall and rate the emotionality of 32 memories. Immediately following Recall1, participants positively re-evaluate half of the memories, and naturally recall the other half. Single-trial multivariate patterns are extracted during Recall1 and Recall2 from the hippocampus, ventral striatum (VS) and ventromedial prefrontal cortex (VMPFC), and correlation distance (1-corr) is used as a measure of Recall1-Recall2 neural pattern dissimilarity. This is done separately for the 16 positive and the 16 control trials, creating one representational dissimilarity matrix (RDM, or rather a vector along the diagonal) per condition per subject. As reported in the supplements, the authors do not find evidence for a significantly greater pattern change after positive re-evaluation than after natural recall. However, the change in neural similarity scales (positively) with the change in emotion rating in hippocampus and VS, as shown by a second-order analysis correlating the neural pattern change vector with a behavioural emotion change vector. More specifically, the correlation between neural change and emotion change is significantly larger for positive than control trials.

1. The null result in the positive reinterpretation condition compared to the control condition is problematic and makes it difficult to interpret the differential second-order correlations. Surely, if the argument is that the behavioural changes shown in Experiments 1-4 are produced by a change in neural memory traces, then we would expect positive trials to show a greater on-average pattern change than control trials. Without such an overall increase in dissimilarity, it is unclear what is driving the differential correlations with emotion change, especially given that the differences are statistically quite small, and the correlations are based on only 16 trials per participant and might thus be unstable. To make this a more constructive comment, maybe the authors could at least present evidence that those memories that show the most pronounced emotion change (e.g. based on a split-half) do exhibit a significant neural pattern change from Recall1 to Recall2. Also, is it possible that the authors were looking for such pattern changes in the wrong set of brain regions? A whole-brain searchlight using a model matrix of larger pattern change in the positive than the control condition could answer this question.

Reply to Reviews for: NCOMMS-20-34250A-Z

We appreciate that the reviewer brought up this issue. The reason we did not focus on an overall neural dissimilarity difference (detached from emotional change) across the positive and control conditions was because we expect memories to have some degree of change each time they are retrieved, given the reconstructive nature of memory, which has been demonstrated in previous fMRI studies (e.g., Xue et al., 2010). Therefore, evidence of an overall dissimilarity difference (detached from emotion change) may not necessarily indicate a change linked to our manipulation (positive reinterpretation). In addition to this, since not all memories change after using positive meaning finding and those that do change, change to varying degrees, we reasoned that Recall1-Recall2 neural dissimilarity tracking increases in positive emotion would be more meaningful than comparing across positive and control conditions alone.

As suggested by the reviewer, we median split the positive condition and re-ran the analysis examining neural dissimilarity across retrievals between conditions (detached from emotion). The median feeling change was +1 in the positive condition, meaning the most pronounced memories were feeling change $\geq +2$. However, this only comprised about ~19% of trials in the positive condition, and 6 participants did not have any memories that met this criteria (as +1 is a significant proportion of feeling change scores within the positive condition). Therefore, we are fairly underpowered to conduct this analysis. Nevertheless, neural dissimilarity between positive and control conditions was in the expected direction in both the hippocampus ($t_{25} = 1.63$, $p = 0.11$) and NAcc ($t_{25} = 0.75$, $p = 0.46$) but non-significant. Although these results are not significant, and perhaps not surprisingly given its underpowered, it is quite promising that even with a very small number of trials we are seeing these results in the expected direction between positive and control conditions. Echoing our previous point, we would not necessarily expect there to be a difference without considering our behavioral effect, given that we already know that memories change to some degree each time they are retrieved. It is possible that with a larger sample, an analysis of the memories with the most pronounced feeling change could indeed come out. We did conduct a whole-brain searchlight analysis (that we report as an exploratory analysis in the Supplement), but this did not reveal significant activation. Given that memories do change to some degree each time they are retrieved, this additional 'noise' makes it challenging, and we would argue also less meaningful, to focus on changes in neural dissimilarity across conditions in absence of the behavioral effect.

To address the reviewer's point, we have provided our rationale for our key hypothesis linking neural dissimilarity to emotion change in the results section of Exp 4.

Manuscript excerpt (pg 13):

We reasoned that Recall1-Recall2 neural dissimilarity tracking increases in positive emotion would be more meaningful than comparing across positive and control conditions alone since not all memories will necessarily change after using positive meaning finding and those that do change, change to varying degrees¹⁵ (see Supplementary Information for exploratory analyses examining overall neural dissimilarity and exploratory ROIs, e.g., amygdala).

Reply to Reviews for: NCOMMS-20-34250A-Z

2. The results section is also missing statistics to test if the second-order correlation is significant in the positive condition itself (i.e., when not comparing it to the control condition). From Fig. 4 it appears that in the hippocampus, the differential correlation between conditions is produced by the expected positive correlation in the positive condition; however, in the striatum, the difference is mainly produced by a negative correlation in the control condition (see Fig. 4), suggesting that it is the memories with little emotion change that become more dissimilar (or vice versa). Again, the differences between conditions seem meaningless without an overall pattern change, and a significant positive correlation, in the positive condition.

Similar to our reply above for #1, since memory is reconstructed at the time of retrieval, we wouldn't necessarily expect natural recollection to look exactly the same each time it is recalled (e.g, slope of '0'), which has been demonstrated in the memory literature. In addition, the pattern change relationship with emotion change for natural recall may differ depending on the specific ROI. In the 2 regions we highlight in the manuscript, for instance, natural recall memory patterns do have a slope close to 0 in the hippocampus but show a negative relationship with emotion change in the NAcc. Therefore, showing that the positive condition differs in and of itself from 0 isn't necessarily meaningful because 0 isn't necessarily ground truth. Ground truth more likely reflects what happens when we naturally recall a memory, which is why our key analysis sought to compare our positive manipulation with natural recall, and the degree to which it changed as a function of emotion. As suggested by the reviewer, when we test the positive condition against zero with a one-sample t-test, neural dissimilarity significantly differs from zero in the hippocampus ($t_{31} = 2.82$, $p = .008$) and is in the expected direction but non-significant in the NAcc ($t_{31} = 1.54$, $p = .133$). We now report these analyses in our Supplement.

Supplement excerpt (pg 12):

We also tested whether neural dissimilarity across retrievals in the positive condition alone was significantly different from zero in our two ROIs that showed a significant difference between positive and control conditions. Neural dissimilarity significantly differed from zero in the hippocampus ($t_{31} = 2.82$, $p = .008$) and is in the expected direction but non-significant in the NAcc ($t_{31} = 1.54$, $p = .133$).

3. Finally, the univariate differences (and their absence in some contrasts), reported in the supplements, are not entirely convincing. The Recall2 positive > control contrast shows no significant difference between conditions. Differences during the elaboration task are found in regions that might reflect generally higher effort (VLPFC, DLPFC), and not necessarily emotion regulation in particular.

Overall, the fMRI data from Experiment 4 are the least compelling part in an otherwise coherent series of experiments.

We did not necessarily expect to find significant activation for positive > control contrast during Recall 2 at the whole-brain level as a univariate analysis may not be sensitive enough to detect differences. This is exactly why our key analysis was to examine changes in patterns of activity

Reply to Reviews for: NCOMMS-20-34250A-Z

rather than 'amount' of activation in a particular region between conditions. While the VLPFC and DLPFC regions are multi-functional and have been linked to several processes, one key process is certainly emotion regulation. This has been seen across several experiments and meta-analysis as the VLPFC and DLPFC are the two most commonly observed regions in emotion regulation studies (Ochsner, Silvers, & Buhle, 2012; Wager et al., 2008), and for reappraisal in particular (Buhle et al., 2014; Denny, Inhoff, Zerubavel, Davachi, & Ochsner, 2015), which is very similar to the positive reinterpretation manipulation we used. Indeed, emotion regulation is considered an effortful mental process, and so these regions are certainly involved in effort as well.

Other comments and questions:

4. The statement that data are inconsistent with concurrent or competitive memory traces (see p.15 bottom) seems unfounded and needs to be explained. In my view, most data points reported in the manuscript (except maybe Exp. 3, Immediate Group) are consistent with the idea that participants create an alternative, more positive version of the original memory during the elaboration task, and are then more likely to retrieve this positive re-interpretation during the delayed recall.

To clarify, by stating that our data is inconsistent with concurrent or competitive memory traces, we are not suggesting that participants do not generate a more positive interpretation at all. What our data from Exp 3 suggests is that participants do in fact generate a positive interpretation during the elaboration period. But at future retrieval (24h later), they are not recalling a separate/alternative memory trace (created during elaboration), but rather an updated memory trace that is integrated with original details plus new details from the elaboration. This updated memory is not considered a concurrent or competing memory trace (with the original memory trace in the memory literature). Specifically, our results from Exp 3 show that participants only show increased positive emotion at future retrieval when positive reinterpretation occurred during the reconsolidation window and when retrieval occurred 24h later (i.e., the delayed test group). If it were the case that participants were simply recalling a concurrent or competing memory at future retrieval (that is truly separate from the original memory trace), then we should have observed a memory change in the No-Reminder group (as they were given the opportunity to generate a separate memory outside of the reconsolidation window), but we did not observe this. Therefore, our results are inconsistent with a concurrent or competitive memory trace explanation for observed changes in memory. Importantly, we use the same or a variation of the same paradigm (mirroring the delayed test group in Exp 3) across all studies, therefore we would assume there to be similar updating mechanisms across studies. We now unpack this finding in the discussion section to clarify what we mean by our data being inconsistent with concurrent or competitive memory traces.

Manuscript excerpt (pg 16):

This conceptualization differs from alternative mechanisms that could also lead to future memory change, such as concurrent or competing retrieval of dual memories²², which was not supported by our data (see Exp 3). That is, updates were only observable after a previous

Reply to Reviews for: NCOMMS-20-34250A-Z

reminder during the reconsolidation window (manipulation occurring ~10 min to 6h after reactivation) and a memory test after a sufficient delay (24h later). Importantly, we assume there to be similar mechanisms across studies given that all other paradigms mirrored a version of the *Delayed-test* group in Exp 3.

5. Were trials included in the behavioural and fMRI analyses where subjects indicated they could not think of a positive aspect? I might have missed this information in the methods section, if not it should be clarified.

We do not include trials for which participants were unable to use the positive meaning finding strategy. We have now clarified this in our methods section.

6. Experiment 1: Was the greater increase in positive feeling also correlated with more positive content and dissimilarity in other groups, or was it specific to Positive Group?

Correlations of positive feeling with positive content and dissimilarity were non-significant in the other 3 groups (Negative: $t_{23} = 0.26$, $p = .799$; $t_{23} = -1.28$, $p = .213$; Neutral: $t_{23} = 1.40$, $p = .174$; $t_{23} = -1.50$, $p = .148$; Distraction: $t_{24} = -0.05$, $p = .962$; $t_{24} = 0.76$, $p = .456$). Since this was not one of our predictions, we did not initially examine this. However, we appreciate the suggestion and have now added these exploratory analyses to our Supplement.

Supplement excerpt (pg 4):

Correlations of positive feeling with positive content and dissimilarity were non-significant in the Negative ($t_{23} = 0.26$, $p = .799$; $t_{23} = -1.28$, $p = .213$), Neutral ($t_{23} = 1.40$, $p = .174$; $t_{23} = -1.50$, $p = .148$) and Distraction groups ($t_{24} = -0.05$, $p = .962$; $t_{24} = 0.76$, $p = .456$).

7. Experiment 2: Statistics for the Group x Time ANOVAs missing.

We have added the full statistics for the group x time anovas to Exp 2.

Manuscript excerpt (pg 8):

We examined changes in feeling ratings and memory content (positivity, dissimilarity) via Time (1-week, 2-months) by group (*Positive, Control*) ANOVAs, which yielded significant main effects of time ($F_{1,178} = 11.82$, $p < .001$, $\eta^2 = .023$; $F_{1,178} = 7.28$, $p = .008$, $\eta^2 = 0.037$; $F_{1,178} = 21.31$, $p < .001$, $\eta^2 = 0.098$) and group ($F_{1,178} = 4.43$, $p = .037$, $\eta^2 = .061$; $F_{1,178} = 9.10$, $p = .003$, $\eta^2 = 0.047$; $F_{1,178} = 19.83$, $p < .001$, $\eta^2 = 0.091$), but no interactions ($p = .824$; $p = .840$; $p = .644$).

8. Experiment 4: The paired t-tests used to compare mean correlation coefficients across conditions need to be described in more detail, e.g., what was being permuted/held constant, and how were t-values computed from the empirical null distributions.

We apologize for the lack of clarity about this analysis. We have revised the results and methods sections with more details. We used a one-sample sign permutation test (5000

Reply to Reviews for: NCOMMS-20-34250A-Z

iterations) for the difference in rho values between conditions (positive, control) using the *nitools* python toolbox.

Manuscript excerpt (pg 14-15):

We then compared the mean correlations between conditions (positive, control) at the group level by running a one-sample sign permutation test (5000 iterations) on the difference in rho values between conditions using the *nitools* python toolbox.

9. Also regarding the fMRI analyses (Exp. 4), the authors should explain why a 5mm smoothing kernel was applied before subjecting data to the RSA analysis. RSA/multivariate pattern analyses are typically conducted on unsmoothed data.

While the reviewer is correct that it is more typical to use unsmoothed data, the literature is somewhat mixed when considering the size of your ROIs (Dimsdale-Zucker & Ranganath, 2018). We chose to smooth our data because our ROIs were sufficiently large such that smoothing may improve the signal-to-noise ratio for our pattern similarity analyses, as described in previous research. If we were only examining smaller ROIs, we likely would not have smoothed, as this would have diminished the signal-to-noise ratio. We now provide this justification in the methods section for Exp 4.

Manuscript excerpt (pg 33):

We chose to apply a 5mm smoothing kernel to our data because our ROIs were sufficiently large such that smoothing may improve the signal-to-noise ratio⁵⁸.

Reply to Reviews for: NCOMMS-20-34250A-Z

References

- Buhle, J. T., Silvers, J., Wager, T. D., Lopez, R., Onyemekwu, C., Kober, H., ... Ochsner, K. N. (2014). Cognitive reappraisal of emotion: A meta-analysis of human neuroimaging studies. *Cerebral Cortex*, 24(11), 2981–2990. <https://doi.org/10.1093/cercor/bht154>
- Denny, B. T., Inhoff, M. C., Zerubavel, N., Davachi, L., & Ochsner, K. N. (2015). Getting over it: Long-lasting effects of emotion regulation on amygdala response. *Psychological Science*, 26(9), 1377–1388. <https://doi.org/10.1177/0956797615578863>
- Dimsdale-Zucker, H., & Ranganath, C. (2018). Chapter 27 - Representational Similarity Analyses: A Practical Guide for Functional MRI Applications. In *Handbook of Behavioral Neuroscience* (pp. 509–525). Retrieved from <https://www.sciencedirect.com/science/article/pii/B9780128120286000276?via%3Dihub>
- Dore, B. P., Boccagno, C., Burr, D., Hubbard, A., Long, K., Weber, J., ... Ochsner, K. N. (2016). Finding Positive Meaning in Negative Experiences Engages Ventral Striatal and Ventromedial Prefrontal Reward Regions. *Journal of Cognitive Neuroscience*. <https://doi.org/10.1162/jocn>
- Else, J. W. B., Van Ast, V. A., & Kindt, M. (2018). Human memory reconsolidation: A guiding framework and critical review of the evidence. *Psychological Bulletin*, 144(8), 797–848. <https://doi.org/10.1037/bul0000152>
- Ochsner, K. N., Silvers, J. A., & Buhle, J. T. (2012). Functional imaging studies of emotion regulation: A synthetic review and evolving model of the cognitive control of emotion. *Annals of the New York Academy of Sciences*, 1251, E1-24. <https://doi.org/10.1111/j.1749-6632.2012.06751.x>
- Sevenster, D., Beckers, T., & Kindt, M. (2013). Prediction error governs pharmacologically induced amnesia for learned fear. *Science*, 339, 830–833. <https://doi.org/10.1126/science.1229223>
- St Jacques, P. L., Olm, C., & Schacter, D. L. (2013). Neural mechanisms of reactivation-induced updating that enhance and distort memory. *Proceedings of the National Academy of Sciences of the United States of America*, 110(49), 19671–19678. <https://doi.org/10.1073/pnas.1319630110>
- Wager, T. D., Davidson, M. L., Hughes, B. L., Lindquist, M. A., & Ochsner, K. N. (2008). Prefrontal-subcortical pathways mediating successful emotion regulation. *Neuron*, 59(6), 1037–1050. <https://doi.org/10.1016/j.neuron.2008.09.006>
- Xue, G., Dong, Q., Chen, C., Lu, Z., Mumford, J. A., & Poldrack, R. a. (2010). Greater neural pattern similarity across repetitions is associated with better memory. *Science*, 330(6000), 97–101. <https://doi.org/10.1126/science.1193125>

REVIEWER COMMENTS

Reviewer #1 (Remarks to the Author):

The authors have sufficiently addressed my prior concerns. Just one small comment:

- In Experiment 1 results (p8), the wording was changed to report the main effects and interaction result. However, the following sentence reports only the group main effect. What does the main effect of time show? While this may not be crucial to the story, it may still be worth reporting. In fact, from the figure, it seems as though dissimilarity and content change increase from 1 week to 2 months, which is fascinating.

Reviewer #2 (Remarks to the Author):

The authors have adequately addressed my concerns.
I look forward to seeing this paper published.

Reviewer #3 (Remarks to the Author):

The authors have mostly addressed my comments and concerns. However, there are a few remaining suggestions and points I would like clarified.

1. In additional analyses included in the supplement, the authors report results for hippocampus and NAcc (nucleus accumbens). However, the rest of the paper uses ventral striatum (not NAcc), so the authors may want to stick to using VS for consistency.
2. This is something I didn't notice the first time around, but my attention was drawn to it by reviewer 5's question about smoothing. Did you apply smoothing twice -- first a kernel of 5mm during SPM12 preprocessing, then a 2mm kernel using SUSAN? And if so, can you please provide a justification for this?
3. I'm still confused by the percentages of details reported on page 9 (Exp 2). For example, at 1 week, 74.6% were old details, 10.2% were details from elaboration, 12.2% new related positive details, and 10.9% new negative details (equaling 107.9%). Also, I think it would be informative to include information about the number of details per memory during Recall 2 in each group.
4. Did Exp 4 also have the 10 min delay between recall and elaboration?
5. On page 18, the authors write "fitting with our observation that natural recollection had a stable hippocampal pattern." Which result supports this statement? The main Exp 4 result for the control condition is that there was no correlation between feeling change and neural dissimilarity across recalls, but this doesn't necessarily mean the hippocampal pattern was stable. In the supplement (page 12), neural dissimilarity significantly differed from zero in the hippocampus in the positive condition alone, but the same analysis is not reported for the control condition.
6. Response to reviewer 2, question 10: I do not see the added description to the caption for figure 3A.
7. Response to reviewer 5, question 1: The reviewer suggested a whole-brain searchlight analysis looking for larger pattern change in the positive compared to the control condition. The authors state that they did conduct a whole-brain searchlight analysis that is reported as an exploratory analysis in the Supplement, but I do not see these results in the supplement.
8. Response to reviewer 5, question 5: I do not see the information about excluded trials for experiment 4. In case I somehow overlooked this, can you please point me to where this information is in the text? Were any memories excluded in Exp 4, or did all subjects have exactly 32 memories included in the analysis (16 positive, 16 control)?

9. Response to reviewer 5, question 6: I think it would be helpful to include the non-significant correlation results for the negative, neutral and distraction groups in the main text.

10. Can you please add slice information to supplementary Fig 2?

Reviewer #4 (Remarks to the Author):

My recommendation would be to accept this paper.

Reviewer #5 (Remarks to the Author):

The authors present a thorough revision of the manuscript, incorporating changes in response to all concerns raised in my previous review, and, as far as I could see, the other reviewers too. There were some points where the authors disagreed for good reason (e.g. my previous point #1), and make a convincing point in their response letter why an overall change in neural similarity is not necessarily expected. Also, the manuscript's main scientific message ultimately does not hinge on the details of the fMRI study, because the fMRI data merely provide additional support for the updating mechanism that is convincingly shown in Experiments 1-3.

Overall, the revision addresses all major concerns, and I have no further suggestions.

Reply to Reviews for: NCOMMS-20-34250B

We would like to thank the editor and the reviewers for their time re-reviewing our manuscript after revision. We believe the manuscript has benefitted tremendously from the constructive and detailed feedback from all 5 reviewers. Below, we present a point-by-point response to each of the remaining concerns raised by reviewers 1 and 3. We appreciate the opportunity to provide additional methodological and analysis details that ultimately improve clarity of the text and strengthen the paper. All revisions to manuscript text are included below for convenience but also depicted in the manuscript by red font. Thank you again for your time and helpful feedback.

REVIEWER COMMENTS

Reviewer #1 (Remarks to the Author): The authors have sufficiently addressed my prior concerns. Just one small comment:

- In Experiment 1 results (p8), the wording was changed to report the main effects and interaction result. However, the following sentence reports only the group main effect. What does the main effect of time show? While this may not be crucial to the story, it may still be worth reporting. In fact, from the figure, it seems as though dissimilarity and content change increase from 1 week to 2 months, which is fascinating.

We agree with the reviewer that the time variable in Exp 2 (reported on pg 8) is an interesting finding. We now elaborate on this in the main text.

Manuscript excerpt (pg 8): We examined changes in feeling ratings and memory content (positivity, dissimilarity) via Time (1-week, 2-months) by group (*Positive*, *Control*) ANOVAs, which yielded significant main effects of time ($F_{1,178} = 11.82, p < .001, \eta^2 = .023$; $F_{1,178} = 7.28, p = .008, \eta^2 = 0.037$; $F_{1,178} = 21.31, p < .001, \eta^2 = 0.098$) and group ($F_{1,178} = 4.43, p = .037, \eta^2 = .061$; $F_{1,178} = 9.10, p = .003, \eta^2 = 0.047$; $F_{1,178} = 19.83, p < .001, \eta^2 = 0.091$), but no interactions ($p = .824$; $p = .840$; $p = .644$). **There was an increase in all three variables after 1-week and 2-months, regardless of group, potentially mirroring the fading of negative feelings that naturally occurs with time¹.** Consistent with our key hypothesis, the *Positive* group had a greater increase in positive emotion, positive content, and dissimilarity in event details than the *Control* group...

Reviewer #2 (Remarks to the Author): The authors have adequately addressed my concerns. I look forward to seeing this paper published.

Reviewer #3 (Remarks to the Author): The authors have mostly addressed my comments and concerns. However, there are a few remaining suggestions and points I would like clarified.

1. In additional analyses included in the supplement, the authors report results for hippocampus and NAcc (nucleus accumbens). However, the rest of the paper uses

Reply to Reviews for: NCOMMS-20-34250B

ventral striatum (not NAcc), so the authors may want to stick to using VS for consistency.

We have revised the supplement to state VS rather than NAcc for consistency.

2. This is something I didn't notice the first time around, but my attention was drawn to it by reviewer 5's question about smoothing. Did you apply smoothing twice -- first a kernel of 5mm during SPM12 preprocessing, then a 2mm kernel using SUSAN? And if so, can you please provide a justification for this?

Thank you for pointing this out. We only used the 5mm kernel with SPM12 preprocessing. We removed the text about smoothing with SUSAN.

3. I'm still confused by the percentages of details reported on page 9 (Exp 2). For example, at 1 week, 74.6% were old details, 10.2% were details from elaboration, 12.2% new related positive details, and 10.9% new negative details (equaling 107.9%). Also, I think it would be informative to include information about the number of details per memory during Recall 2 in each group.

We apologize for the confusion. These numbers refer to percentages from different timepoints (recall 1 vs. future recollections) and therefore would not add up to 100%. The first analysis we describe is showing that a majority percentage of details from recall 1 itself are preserved across timepoints, and this is similar across groups. The goal was to demonstrate that people are, in fact, generally recalling the same memory across time rather than a brand new memory. The next set of analyses describe percentages of the future recollections (recall 2, recall 3), such as the % of details that were from the elaboration period, new positive details, or new negative details (that were not present during recall 1). Here, we did not report the percentage of old details included in recall 2 or 3. We now include these percentages along with the number of details per memory during recall 2 and 3 by group, as requested. We hope this improves clarity on this analysis.

Manuscript excerpt (pg 9-10): Groups did not differ in the number of details they recalled per memory after 1-week ($M_{\text{Positive}}=4.19$, $SD=1.37$; $M_{\text{Control}}=3.50$, $SD=2.58$; $t_{89}=1.58$, $p=.117$, $d=0.331$) or 2-months ($M_{\text{Positive}}=4.01$, $SD=1.22$; $M_{\text{Control}}=4.04$, $SD=1.10$; $t_{89}=-0.11$, $p=.912$, $d=0.023$). The *Positive* group's future recollections included about 10.2% of their positive elaboration at 1-week and 10.5% at 2-months ($t_{45}=4.77$, $p<.001$; $t_{45}=5.27$, $p<.001$). Their future recollections also included 12.2% and 14.1% of new related positive details at 1-week and 2-months respectively, which was significantly greater than the *Control* group (1-week: 8.1%, $t_{89}=2.63$, $p=.01$, $d=0.55$; 2-months: 10.2%, $t_{89}=2.29$, $p=.025$, $d=0.48$; percentage of new details corresponds to the number of new details divided by the total number of details during retrieval). Groups did not differ in the number of new negative details that were present at 1-week ($M_{\text{Positive}}=10.9\%$, $M_{\text{Control}}=8.3\%$; $t_{89}=1.74$, $p=.085$, $d=0.37$) or 2-months ($M_{\text{Positive}}=10.7\%$, $M_{\text{Control}}=8.8\%$; $t_{89}=1.42$, $p=.16$, $d=0.30$). Given that the *Positive* group incorporated aspects of their positive elaboration and more new positive details than the *Control* group, a smaller

Reply to Reviews for: NCOMMS-20-34250B

percentage of their future recollections included initial details from Recall 1 (1-week: $M_{\text{Positive}}=66.7\%$, $M_{\text{Control}}=79.8\%$; $t_{89}=-3.75$, $p=.003$, $d=0.788$; 2-months: $M_{\text{Positive}}=64.7\%$, $M_{\text{Control}}=78.4\%$; $t_{89}=-4.00$, $p=.0001$, $d=0.841$).

4. Did Exp 4 also have the 10 min delay between recall and elaboration?

Yes, Exp 4 follows the same protocol as the delayed test group in Exp 3 that had a 10 min delay between recall and elaboration. We added a reminder of this detail to Exp 4.

Manuscript excerpt (pg 13): To test this, Experiment 4 followed a modified version of the *Delayed-test* group in Experiment 3, including the 10-min delay between recall and elaboration.

5. On page 18, the authors write “fitting with our observation that natural recollection had a stable hippocampal pattern.” Which result supports this statement? The main Exp 4 result for the control condition is that there was no correlation between feeling change and neural dissimilarity across recalls, but this doesn’t necessarily mean the hippocampal pattern was stable. In the supplement (page 12), neural dissimilarity significantly differed from zero in the hippocampus in the positive condition alone, but the same analysis is not reported for the control condition.

We used the word ‘stable’ to refer to it being highly similar (regardless of emotion change) across time, given no significant pattern change. However, we are happy to modify the terminology in our paper to state more ‘similar’ activation patterns in the control condition rather than ‘stable’, given that our analysis was examining dissimilarity. We have revised this sentence.

Manuscript excerpt (pg 19): Importantly, specific episodic details are distinguishable across multivariate patterns within the hippocampus², fitting with our observation that natural recollection had a similar hippocampal pattern, whereas positive meaning finding had greater pattern dissimilarity that tracked increases in positivity across time.

6. Response to reviewer 2, question 10: I do not see the added description to the caption for figure 3A.

The added description for figure 3A was updated in the figure captions at the very end of the manuscript. However, by mistake, we did not update this in the figure shown within the manuscript text. We have revised this (see pg 12).

7. Response to reviewer 5, question 1: The reviewer suggested a whole-brain searchlight analysis looking for larger pattern change in the positive compared to the control condition. The authors state that they did conduct a whole-brain searchlight analysis that is reported as an exploratory analysis in the Supplement, but I do not see these results in the supplement.

Reply to Reviews for: NCOMMS-20-34250B

We apologize for this omission. The null results of the whole-brain searchlight analysis are now reported in the Supplement.

Supplement excerpt (pg 12): We ran our key RSA analysis as an exploratory whole-brain searchlight RSA analysis using PyMVPA. This analysis performed a searchlight across the brain revealing regions where the correlation between Recall1-Recall2 dissimilarity and increased positivity across time was greater in the positive condition relative to the control condition. We used the same inputs (parameter estimates from a single-trial memory first-level GLM) as the ROI-based RSA analyses. We corrected for multiple comparisons via a non-parametric permutation test (5000 iterations) to reach a corrected alpha < .05. However, this analysis yielded no significant activation after correction.

8. Response to reviewer 5, question 5: I do not see the information about excluded trials for experiment 4. In case I somehow overlooked this, can you please point me to where this information is in the text? Were any memories excluded in Exp 4, or did all subjects have exactly 32 memories included in the analysis (16 positive, 16 control)?

We used the same exclusion criteria for Exp 4 as we did for all experiments, which is described in the Exclusion criteria and data analysis common to all Experiments section of our methods. We added an additional sentence to the methods section of Exp 4 to report the percentage of excluded trials.

Manuscript excerpt (pg 32-33): This analysis only included memories that met criteria (as described in the exclusion criteria section and common to all experiments; 23.1% of trials were excluded).

9. Response to reviewer 5, question 6: I think it would be helpful to include the non-significant correlation results for the negative, neutral and distraction groups in the main text.

As requested, we moved the non-significant correlation results for the negative, neutral and distraction groups from the supplement to the main text.

Manuscript excerpt (pg 6): Correlations of positive feeling with positive content and dissimilarity were non-significant in the *Negative* ($r_{23} = .054$, $p = .799$; $r_{23} = -.258$, $p = .213$), *Neutral* ($r_{23} = .281$, $p = .174$; $r_{23} = -.298$, $p = .148$) and *Distraction* groups ($r_{24} = -.010$, $p = .962$; $r_{24} = .153$, $p = .456$).

10. Can you please add slice information to supplementary Fig 2?

We have added slice information to supplementary Fig 2 (see Supplement page 11).

Reply to Reviews for: NCOMMS-20-34250B

Reviewer #4 (Remarks to the Author): My recommendation would be to accept this paper.

Reviewer #5 (Remarks to the Author): The authors present a thorough revision of the manuscript, incorporating changes in response to all concerns raised in my previous review, and, as far as I could see, the other reviewers too. There were some points where the authors disagreed for good reason (e.g. my previous point #1), and make a convincing point in their response letter why an overall change in neural similarity is not necessarily expected. Also, the manuscript's main scientific message ultimately does not hinge on the details of the fMRI study, because the fMRI data merely provide additional support for the updating mechanism that is convincingly shown in Experiments 1-3.

Overall, the revision addresses all major concerns, and I have no further suggestions.

References

1. Ritchie, T. D. *et al.* A pancultural perspective on the fading affect bias in autobiographical memory. *Memory* **23**, 278–290 (2015).
2. Chadwick, M. J., Hassabis, D., Weiskopf, N. & Maguire, E. A. Decoding Individual Episodic Memory Traces in the Human Hippocampus. *Curr. Biol.* **20**, 544–547 (2010).

REVIEWERS' COMMENTS

Reviewer #3 (Remarks to the Author):

The authors have adequately addressed my concerns. I have one remaining suggestion/point of clarification, but I trust the authors will address it and I do not need to see the paper again.

In response to my prior point #5, I assumed that 'stable' referred to the hippocampal pattern being highly similar across retrievals in the control condition (regardless of emotion change). However, I do not see that specific result reported in the manuscript. On page 12 of the supplement, the authors test whether neural dissimilarity across retrievals was significantly different from zero, but only test this and report results for the positive condition. I suggest the authors also report the overall neural dissimilarity across retrievals for the control condition (which presumably does not differ from zero in the hippocampus).

Reply to reviewers for NCOMMS-20-34250C

We thank reviewer #3 and the editorial staff for re-reviewing our paper. We have addressed the remaining comment from reviewer #3.

REVIEWERS' COMMENTS

Reviewer #3 (Remarks to the Author):

The authors have adequately addressed my concerns. I have one remaining suggestion/point of clarification, but I trust the authors will address it and I do not need to see the paper again.

In response to my prior point #5, I assumed that 'stable' referred to the hippocampal pattern being highly similar across retrievals in the control condition (regardless of emotion change). However, I do not see that specific result reported in the manuscript. On page 12 of the supplement, the authors test whether neural dissimilarity across retrievals was significantly different from zero, but only test this and report results for the positive condition. I suggest the authors also report the overall neural dissimilarity across retrievals for the control condition (which presumably does not differ from zero in the hippocampus).

Reply: We have now added the t-test examining whether neural dissimilarity in the hippocampus across retrievals is significantly different than zero for the control condition, which is non-significant.

Excerpt from Supplementary Information (pg. 13): For the control condition alone, neural dissimilarity across retrievals did not significantly differ from zero in the hippocampus ($t_{31} = -0.30$, $p = .765$).